

# Different Pathways of the Formation of Highly Oxidized Multifunctional Organic Compounds (HOMs) from the Gas-Phase Ozonolysis of β-Caryophyllene

Stefanie Richters, Hartmut Herrmann, Torsten Berndt

Leibniz Institute for Tropospheric Research, TROPOS, D-04315 Leipzig (Germany)

*Correspondence to*: Stefanie Richters (richters@tropos.de)

**Abstract.** The gas-phase mechanism of the formation of highly oxidized multifunctional organic compounds (HOMs) from the ozonolysis of β-caryophyllene was investigated in a free-jet flow system at atmospheric pressure and a temperature of $295 \pm 2$ K. Reaction products, mainly highly oxidized $RO_2$ radicals, containing up to 14 oxygen atoms were detected using

chemical ionization – atmospheric pressure interface – time-of-flight mass spectrometry with nitrate and acetate ionization. These highly oxidized $RO_2$ radicals react with NO, $NO_2$, $HO_2$ and other $RO_2$ radicals under atmospheric conditions forming the first-generation HOM closed-shell products.

Mechanistic information on the formation of the highly oxidized $RO_2$ radicals are based on results obtained with isotopically labeled ozone ($^{18}O_3$) in the ozonolysis reaction and from H/D exchange experiments of acidic H atoms in the products. The

experimental findings indicate that HOM formation in this reaction system is considerably influenced by the presence of a double bond in the $RO_2$ radicals primarily formed from the β-caryophyllene ozonolysis. Three different reaction types for HOM formation can be proposed allowing to explain the detected main products, i.e. i) the simple autoxidation, corresponding to the repetitive reaction sequence of intramolecular H-abstraction of a $RO_2$ radical, $RO_2 \rightarrow QOOH$, and subsequent $O_2$ addition forming a next peroxy radical, $QOOH + O_2 \rightarrow R´O_2$, ii) an extended autoxidation mechanism additionally involving the internal

reaction of a $RO_2$ radical with a double bond forming most likely an endoperoxide, and iii) an extended autoxidation mechanism including $CO_2$ elimination. The individual reaction steps of the reaction types ii) and iii) are uncertain at the moment. From the product analysis it can be followed that the simple autoxidation mechanism accounts only for about one third of the formed HOMs.

Time-dependent measurements showed that the HOM formation proceeds at a timescale of 3 seconds or less under the

concentration regime applied here.

The new reaction pathways represent an extension of the mechanistic understanding of HOM formation via autoxidation in the atmosphere, as recently discovered from laboratory investigations on monoterpene ozonolysis.



# 1 Introduction

The emission of biogenic volatile organic compounds (BVOCs) from vegetation to the troposphere and their oxidation in the gas phase is subject of intense research (Calvert et al., 2000; Guenther et al., 2012; Ziemann and Atkinson, 2012). Sesquiterpenes (SQTs, $C_{15}H_{24}$) with an annual emission of 18-24 million metric tons carbon (Messina et al., 2015; Sindelarova

et al., 2014) contribute with up to 3 % to the annual global BVOC emission of 720-1150 million metric tons of carbon (Guenther et al., 1995; Guenther et al., 2012; Lathière et al., 2005; Sindelarova et al., 2014). They are emitted by a large variety of plants and fungi and their emission pattern depends strongly on the region and the season (Ciccioli et al., 1999; Duhl, 2008; Geron and Arnts, 2010; Horváth et al., 2011; Jardine et al., 2011). Biotic stress can drastically increase SQT emissions (Mentel et al., 2013). β-Caryophyllene emissions were calculated to account for 25 % of global SQT emissions (Guenther et al., 2012)

and can contribute 70 % to the regional BVOC emissions, e.g. in orange orchards (Ciccioli et al., 1999; Duhl, 2008). The oxidation products are expected to have a very low vapor pressure making them important for the process of secondary organic aerosol (SOA) formation (Jaoui et al., 2013; Zhao et al., 2015).

β-Caryophyllene is mainly oxidized by ozone under atmospheric conditions having a lifetime $\tau_{(O_3)} = 2$ min for an average ozone concentration of $[O_3] = 7 \times 10^{11}$ molecules cm$^{-3}$ (Finlayson-Pitts and Pitts, 1986) and a rate coefficient

$k_{(296\ K)} = 1.1 \times 10^{-14}$ cm$^3$ molecule$^{-1}$ s$^{-1}$ (Richters et al., 2015; Shu and Atkinson, 1994). Gas-phase product formation from the ozonolysis of β-caryophyllene was already studied in a series of laboratory investigations (Calogirou et al., 1997; Grosjean et al., 1993; Jaoui et al., 2003; Lee et al., 2006; Winterhalter et al., 2009) and by means of theoretical calculations (Nguyen et al., 2009). A large variety of carbonyl, epoxide and carboxyl compounds containing up to five oxygen atoms were experimentally observed using different detection techniques. The total carbon yield, comprising gas and particle phase products, accounts for

up to 64 % (Jaoui et al., 2003). A summary of available data in the literature is given by Winterhalter et al. (2009). DFT quantum chemical calculations were conducted accompanying the experimental work by Winterhalter et al. (2009) with special attention to the first oxidation steps. The fraction of stabilized Criegee intermediates at atmospheric pressure was calculated to be 74 %, slightly higher than the experimental value of 60 %. Furthermore, the calculations support the proposed uni- and bimolecular reaction pathways of the Criegee intermediates as proposed from the experimental work. The main reaction

product was stated to be the secondary ozonide with a yield of 64 %. The formation of acids should account for 8 %, dominated by the formation of caryophyllonic acid (Nguyen et al., 2009). This value is slightly lower than the overall gas- and aerosol phase yield of 13.5 % for caryophyllonic acid measured by Jaoui et al. (2003).

Recently, Ehn et al. (2012); (2014) detected highly oxidized multifunctional organic compounds (HOMs) from the oxidation of α-pinene in field and laboratory studies. These HOMs contain up to twelve oxygen atoms and are supposed to have a very

low vapor pressure, which led to their classification as extremely low-volatility organic compounds (ELVOCs) and as important candidates for SOA formation (Ehn et al., 2014).

Other experimental work on HOM formation from the ozonolysis of monoterpenes (Jokinen et al., 2014; Mentel et al., 2015) and model substances, such as cyclohexene (Berndt et al., 2015b; Mentel et al., 2015; Rissanen et al., 2014), led to the





development of an autoxidation mechanism based on $RO_2$ radical chemistry. In this process, an $RO_2$ radical internally abstracts an H atom forming an alkyl radical with a hydroperoxide moiety ($RO_2 \rightarrow QOOH$). Subsequent oxygen addition forms a next $R'O_2$ radical ($QOOH + O_2 \rightarrow R'O_2$) (Berndt et al., 2015b; Crounse et al., 2013; Ehn et al., 2014; Jokinen et al., 2014; Rissanen et al., 2014), which can repeat this reaction sequence. The overall process results in a repetitive oxygen insertion into the

5 molecules on a time scale of seconds (Jokinen et al., 2014). The principle of autoxidation is well known from the liquid phase since more than hundred years (Berezin et al., 1996; Jazukowitsch, 1875) and was recently extended to atmospheric gas-phase reactions (Crounse et al., 2013).

For alkenes with multiple double bonds, such as β-caryophyllene, this mechanism can become more complex caused by the variety of possible reaction pathways of unsaturated $RO_2$ radicals formed as the intermediates. A recent study from this

laboratory showed that the HOM formation from the ozonolysis of α-cedrene (a SQT that contains only a single double bond) was completely explainable by the autoxidation mechanism initiated by the ozone attack at the double bond (Richters et al., 2016). On the other hand, in the case of the analogous reaction of β-caryophyllene (containing two double bonds), the product spectrum was more complex and not fully in line with an autoxidation mechanism ($RO_2 \rightarrow QOOH$, $QOOH + O_2 \rightarrow R'O_2$). This fact points to additional reaction pathways for HOM generation most likely caused by the presence of a second double

bond.

The scope of the present work is the mechanistic elucidation of possible, new reaction pathways of HOM formation starting from the ozonolysis of β-caryophyllene. Experiments with heavy water ($D_2O$) and isotopically labeled ozone ($^{18}O_3$) were conducted in order to obtain additional information on elementary reaction pathways needed to explain the observed products. This approach allowed to develop an extended mechanism for the HOM formation from the ozonolysis of β-caryophyllene.

**2 Experimental**

The gas-phase ozonolysis of β-caryophyllene was investigated in a free-jet flow system at a temperature of $295 \pm 2$ K and a pressure of 1 bar purified air. The experimental approach is described in detail in the literature (Berndt et al., 2015a; 2015b; Richters et al., 2016) and only a brief summary will be given here.

Experiments in the free-jet flow system (outer tube: length: 200 cm, 15 cm inner diameter and a moveable inner tube: 9.5 mm

outer diameter with a nozzle) were conducted under conditions of negligible wall-loss of products and with a reaction time of 3.0-7.9 s (Berndt et al., 2015a). The inner flow of 5 L min$^{-1}$ (STP), containing varying ozone concentrations, was injected through a nozzle to the outer air flow of 95 L min$^{-1}$ (STP) which contained β-caryophyllene and $CH_3COOH$ if needed. Turbulent gas mixing downstream the nozzle rapidly generates a homogeneously mixed reactant gas.

Ozone was produced by passing air or $^{18}O_2$, premixed in $N_2$, through an ozone generator (UVP OG-2) and was measured at

30 the outflow of the reactor by a gas monitor (Thermo Environmental Instruments 49C). All gas flows were set by calibrated gas flow controllers (MKS 1259/1179). β-Caryophyllene was stored in flasks maintained at 278 K, carried along with 38-48 cm$^3$





min$^{-1}$ (STP) nitrogen, and diluted with the air stream just before entering the flow system. Gas chromatography with a flame – ionization detector (GC-FID; Agilent 6890) as well as proton transfer reaction – mass spectrometry (PTR-MS; HS PTR-QMS 500, Ionicon) served as the analytical techniques for β-caryophyllene detection.

The absolute β-caryophyllene concentrations were determined using the "effective carbon-number approach" from GC-FID

analysis using a series of reference substances with known concentrations (Scanlon and Willis, 1985). The reference substances were α-pinene, β-pinene and limonene. The ratio of the effective carbon numbers (equal to the signal ratio for identical sample concentrations) of β-caryophyllene with respect to these monoterpenes is 1.5 (Helmig et al., 2003; Scanlon and Willis, 1985). Before each measurement series, the concentration was determined using GC-FID analysis measuring the β-caryophyllene signal as well as the signals of the reference substances with known concentrations simultaneously. The β-caryophyllene

concentration in the flow system was continuously monitored throughout the experiments by PTR-MS measurements following the ion traces at 205, 147 and 137 amu.

The β-caryophyllene conversion was varied by changing the initial ozone concentration for otherwise constant reaction conditions. The needed gas mixture of $CH_3COOH$ was prepared in a gas-mixing unit.

The reactant gases used had the following purities: β-caryophyllene (98.5 %; Aldrich), $CH_3COOH$ (Aldrich; 99.5 %), $N_2$ (Air

Products; 99,9992 %), $^{18}O_2$ (euriso-top, isotopic enrichment 96 %). Air was taken from a PSA (Pressure Swing Adsorption) unit with further purification by activated charcoal and 4Å molecular sieve. If needed, humidified air was produced by passing a part of the air flow through water saturators filled with $D_2O$ (Aldrich, 99.9 atom %).

Reaction products were detected and quantified by means of chemical ionization – atmospheric pressure interface – time-of-flight (CI-APi-TOF) mass spectrometry (Airmodus, Tofwerk) using nitrate ions and acetate ions for chemical ionization. The

mass spectrometer settings as well as the approach applied for the determination of HOM concentrations are equal to those described in detail by Berndt et al. (2015b). All stated concentrations represent lower limits (Berndt et al., 2015b). The calculation of HOM concentrations and information about detection limitations and the mass axis calibration are given in the supplementary information.

The initial concentrations were (unit: molecules cm$^{-3}$): [β-caryophyllene] = (8.3-8.6) x $10^{10}$; [$O_3$] = (4.7-102) x $10^{10}$ and

[$CH_3COOH$] = (0-1.4) x $10^{14}$.

## 3 Results and Discussion

A series of different experiments was conducted in order to investigate the product formation from the ozonolysis of β-caryophyllene in more detail. In Sect. 3.1, three different groups of products are proposed as a result of the identified signals from mass spectra recorded from runs with nitrate and acetate ionization. The experimental findings utilized for the signal

assignment to the different product groups are described in the following sections. Section 3.2 discusses results from experiments with normal ($^{16}O_3$) or isotopically labeled ozone ($^{18}O_3$) that allow to distinguish between the origin of the O-atoms



in the reaction products arising either from attacking ozone or from air-$O_2$. Experiments with $D_2O$ addition in the carrier gas provide information about the total number of acidic H atoms in each reaction product, being equal to the number of OH and OOH groups, see Sect. 3.3.

## 3.1 Three groups of highly oxidized products

Figure 1 shows two product mass spectra from β-caryophyllene ozonolysis in the mass-to-charge range 345-505 Th, recorded a) with acetate ionization and b) with nitrate ionization. The products appear as adducts with the reagent ion (Ehn et al., 2014). Here, a signal of the same product shows a shift by three nominal mass units comparing acetate ion adducts (+59 nominal mass units) with nitrate ion adducts (+62 nominal mass units). Mainly $RO_2$ radicals were detected as reaction products because the $RO_2$ radical concentrations did not exceed 9 x $10^6$ molecules cm$^{-3}$ and bimolecular reactions of the formed $RO_2$ radicals were

less efficient for a reaction time of 3.0-7.9 s in these experiments. Therefore, the discussion is mainly focused on $RO_2$ radicals. The observed product signals were classified in three product groups. The position of the dominant signals in each product group differs by 32 nominal mass units each due to the stepwise insertion of oxygen molecules.

Signals of the first group, the so-called normal autoxidation group, "norm. AutOx." appear at the same positions in the mass spectrum as observed from the HOM formation of α-cedrene ozonolysis (an SQT with only one double bond, but with the

same chemical formula $C_{15}H_{24}$ like β-caryophyllene) (Richters et al., 2016). The $RO_2$ radicals from this group were summarized by the general formula $O,O-C_{15}H_{23-x}(OOH)_xO_2$ with x = 1-5 (Jokinen et al., 2014; Richters et al., 2016). Here, x stands for the number of hydroperoxide moieties in the molecule, the two oxygen atoms "O,O" arise from the initial ozone attack and the final $O_2$ stands for the $RO_2$ radical functional group (Jokinen et al., 2014). The carbon skeleton of 15 carbon atoms is retained and up to 14 oxygen atoms are inserted in the products.

The second product group, the extended autoxidation group "ext. AutOx.", comprises the signals of $RO_2$ radicals with the general formula $O,O-C_{15}H_{23-y}(OO)(OOH)_yO_2$ with y = 1-4. Here, "(OO)" stands - most likely - for an endoperoxide group. Reactions leading to this insertion step are discussed in the reaction mechanisms in Sect. 3.4. $RO_2$ radicals from the "norm. AutOx." group with $O,O-C_{15}H_{23-x}(OOH)_xO_2$ have the same chemical composition, and consequently the same position in the mass spectrum like the $RO_2$ radicals from the "ext. AutOx." group. A distinction is possible measuring the number of acidic

H atoms in the molecules (equal to the number of OOH groups) applying H/D exchange experiments with heavy water (Rissanen et al., 2014). Products of the "ext. AutOx." group contain one acidic H atom less than the corresponding product from "norm. AutOx." with the same composition, for instance for $C_{15}H_{23}O_8$: $O,O-C_{15}H_{23-y}(OO)(OOH)_yO_2$ with y = 1 and $O,O-C_{15}H_{23-x}(OOH)_xO_2$ with x = 2. Up to now, H/D exchange experiments were successfully conducted in order to elucidate the structure of highly oxidized reaction product from the ozonolysis of cyclohexene which represents a model compound for

cyclic monoterpenes (Berndt et al., 2015b; Rissanen et al., 2014). In the case of cyclohexene ozonolysis, the formation of HOMs strictly followed the normal autoxidation mechanism and the results of H/D exchange experiments confirmed the expected number of hydroperoxide moieties in the products.



The third product group (extended autoxidation with $CO_2$ elimination) named "ext. AutOx -$CO_2$", includes the signals of HOMs with a $C_{14}$ skeleton formed by $CO_2$ elimination in the course of their formation. Based on experiments with isotopically labeled ozone ($^{18}O_3$) (Fig. 2) and heavy water (Fig. 4), highly oxidized $RO_2$ radicals of this product group were assigned to the general formula O-$C_{14}H_{23-\alpha}(O)(OOH)_\alpha O_2$ with $\alpha$ = 1-3. Here, only one oxygen atom from the ozone attack "O-" is retained in the HOM. An additional oxygen atom "(O)" is inserted into the molecule arising from air-$O_2$. It is assumed that this "(O)" exists in an epoxide ring. A possible reaction sequence leading to epoxide formation is discussed in Sect. 3.4.

Closed-shell products in all three product groups were detected at minus 17 nominal mass units compared with the position of the respective $RO_2$ radical in the mass spectrum. The formation of closed-shell products as a result of consecutive, uni- or bimolecular reactions of the $RO_2$ radicals can be explained by a formal loss of one oxygen and one hydrogen atom from the $RO_2$ radical, see proposed reaction pathways as given by Jokinen et al. (2014).

The same reaction products ($RO_2$ radicals and closed-shell products) were detected by means of both ionization methods and all signal assignments were supported by the exact mass-to-charge ratio of the signals (resolving power at 393 Th: 4100 Th/Th). The detected signal intensity (normalized by the reagent ion intensity) of the same HOM measured by both ionization techniques is not necessarily identical caused by possible differences of the cluster ion stability (Berndt et al., 2015b; Hyttinen et al., 2015). As a result of our analysis, acetate ionization is more sensitive especially for the detection of HOMs that contain only one hydroperoxide moiety, O,O-$C_{15}H_{23-x}(OOH)_x O_2$ with x = 1 and O-$C_{14}H_{23-\alpha}(O)(OOH)_\alpha O_2$ with $\alpha$ = 1. A similar observation has been already done for reaction products from the ozonolysis of cyclohexene (Berndt et al., 2015b). The signals of the HOMs with only one hydroperoxide moiety dominate the spectrum recorded with acetate ionization (Fig. 1a) but are of minor importance in the case of nitrate ionization (Fig. 1b). Table 1 summarizes the nominal mass-to-charge ratios of the detected signals and their assignments.

The investigation of the signal intensities points to an important role of reaction products from the "ext. AutOx." and "ext. AutOx. -$CO_2$" groups for the total HOM formation from the ozonolysis of β-caryophyllene. The relative contribution of reaction products from the "ext. AutOx." group to the total molar HOM yield, investigated in the presence of $D_2O$ using nitrate ionization, was determined to be 49 %. The "norm. AutOx." group contribute with 29 % and the "ext. AutOx. -$CO_2$" with 22 % to the total molar HOM yield. The change of the detection sensitivity for different HOMs (especially for those containing a single hydroperoxide moiety) leads to a different contribution of the individual product groups to the total molar HOM yield when changing from nitrate ionization to acetate ionization. For acetate ionization, the "ext. AutOx. -$CO_2$" group contributes with 50 %, the "norm. AutOx." group with 35 % and the "ext. AutOx." group only with 15 % to the total molar HOM yield. Thus, the "norm. AutOx." group contributes on average only with 29-35 % to the total molar HOM yield. The two new product groups "ext. AutOx." and "ext. AutOx. -$CO_2$" are crucial for the explanation of HOM formation from the ozonolysis of β-caryophyllene.



## 3.2 Experiments with isotopically labeled ozone ($^{18}O_3$)

The signal assignment of the three reaction product groups was supported by experiments using isotopically labeled ozone, $^{18}O_3$. When changing from $^{16}O_3$ to $^{18}O_3$ in the ozonolysis, the product signals in the mass spectra were shifted by two nominal mass units for each oxygen arising from the initial ozone attack (Jokinen et al., 2014).

For example, Figure 2 shows a comparison of results from an experiment using either $^{18}O_3$ or $^{16}O_3$ in the ozonolysis reaction for otherwise constant reaction conditions. The spectra in the range 340-400 Th are dominated by four signals of $RO_2$ radicals at the nominal mass-to-charge ratio of 346, 358, 378 and 390 Th representing signals of all three product groups. The signals at nominal 358 and 390 Th were shifted by four nominal mass units when changing from $^{16}O_3$ to $^{18}O_3$. This shift indicates the presence of two oxygen atoms in these reaction products from the initial ozone reaction. The signal at nominal 358 Th is

attributed to a $RO_2$ radical from the "norm. AutOx." group, the signal at nominal 390 Th contains contributions from products of the "norm. AutOx." as well as the "ext. AutOx." group. (A further differentiation by means of H/D exchange experiments is described later.) The signal shift by four nominal mass units shows that reaction products from both product groups contain two oxygen atoms from the initial ozone attack, "O,O", as stated in the general formulas O,O-$C_{15}H_{23-x}(OOH)_xO_2$ with x = 1-5 ("norm. AutOx.") and O,O-$C_{15}H_{23-y}(OO)(OOH)_yO_2$ with y = 1-4 ("ext. AutOx."). The third oxygen atom from the attacking

ozone is the oxygen atom of the OH radical that was split-off from the Criegee intermediate forming the alkyl radicals **4a-4c**, see the first steps of the ozonolysis mechanism in Fig. 5.

The signals at nominal 346 and 378 Th were shifted by two nominal mass units comparing the results using either $^{16}O_3$ or $^{18}O_3$. Consequently, only one oxygen atom from the initial ozone attack remains in these reaction products and a second oxygen atom from the initial ozone attack must be abstracted in the course of the product formation. The position and the exact mass-

20 to-charge ratio of these $RO_2$ signals in the mass spectra suggest that the $RO_2$ radicals contain only 14 carbon atoms. The loss of one carbon atom and one more oxygen atom from the initial ozone attack points to an elimination of CO or $CO_2$ in these molecules. The elimination of CO from highly oxidized $RO_2$ radicals was proposed for reaction products from the ozonolysis of cyclohexene (Berndt et al., 2015b). The corresponding reaction products from the ozonolysis of β-caryophyllene including a CO elimination were detected in small yields at nominal 365, 397 and 429 Th using nitrate ionization and were not further

investigated here.

On the other hand, the formation of reaction products from the third product group is supposed to involve a $CO_2$ elimination starting from species **7** in Fig. 7. Species **7** contains an acylperoxy radical functional group which might react with the double bond under formation of an acylalkoxy radical. From this acylalkoxy radical, $CO_2$ can easily be released (Jaoui et al., 2003; Winterhalter et al., 2009). Therefore, the reaction product at nominal 346 Th can be explained by an elimination of $CO_2$ (-44 nominal mass units) and a subsequent $O_2$ addition (+32 nominal mass units). Reaction products with signals at nominal 378

and 420 Th can be formed by further $O_2$ insertion via autoxidation starting from **17**, see Fig. 7. Based on these results, a $CO_2$ elimination was proposed for reaction products from the third product group, named "ext. AutOx. -$CO_2$". Products of this group can be explained by the general formula O-$C_{14}H_{23-\alpha}(O)(OOH)_\alpha O_2$ with α = 1-3. Here, "O-" stands for the remaining



oxygen atom from the reacting ozone. The proposed reaction mechanism for the formation of the first member of the "ext. AutOx. -CO₂" group with α = 1 is given in Fig. 7, **7'** → **15** → **16** → **17**. It includes tentatively the formation of an epoxide ring. The corresponding oxygen atom is marked as "(O)" in the general formula O-$C_{14}H_{23-\alpha}(O)(OOH)_{\alpha}O_2$. The marked oxygen atom "(O)" could also belong to an aldehyde or ketone. However, it was not possible to explain the formation of a carbonyl

functional group together with the $CO_2$ elimination using known reaction mechanisms in the literature (Jaoui et al., 2003; Winterhalter et al., 2009). On the other hand, epoxide formation was already postulated for the OH radical-initiated oxidation of aromatic compounds (Andino et al., 1996; Bartolotti and Edney, 1995; Berndt and Böge, 2006; Ghigo and Tonachini, 1999; Suh et al., 2003). The explanation of the oxygen atom "(O)" by a hydroxide moiety can be excluded, because this would imply the presence of two more hydrogen atoms in the product and hence an increase by two nominal mass units in the mass spectrum.

Furthermore, the possible presence of a hydroxyl moiety would provide an additional acidic H atom in the molecule, which was not detected in heavy water experiments (see Sect. 3.3).

### 3.3 Experiments with heavy water (D₂O)

A next set of experiments was conducted in presence of heavy water (D₂O), applying nitrate ionization, see Fig. 3 and 4. The addition of D₂O leads to an H/D exchange of all acidic H atoms present in the molecule (Rissanen et al., 2014) and thus, to a

15 signal shift in the mass spectrum by a certain number of nominal mass units being equal to the number of acidic H atoms in the molecule. For HOMs following the normal autoxidation process, all oxygen molecules inserted into the molecule, except the RO₂ radical functional group, are present as hydroperoxide moieties. The resulting signal shift in the presence of D₂O corresponds to the number of hydroperoxide moieties as shown for the HOMs from the ozonolysis of cyclohexene (Berndt et al., 2015b; Rissanen et al., 2014) and α-cedrene (Richters et al., 2016).

Figure 3 shows mass spectra in the presence and absence of D₂O focusing on the signals at nominal 393, 408 and 425 Th which were assigned to reaction products of the "norm. AutOx." and "ext. AutOx." groups. The full spectra in the nominal mass-to-charge range 360-495 Th are shown in Fig. S1. In presence of D₂O, all three signals were split up into two signals according to their numbers of acidic H atoms in the molecules. This behavior indicates that two different reaction products contribute to each signal. The signal at nominal 393 Th corresponds to the RO₂ radical $C_{15}H_{23}O_8$ and was shifted by one or two nominal

mass units when adding D₂O. Two of the eight oxygen atoms arise from the initial ozone attack (see Sect. 3.2) and two oxygen atoms represent the RO₂ radical functional group. Consequently, two oxygen molecules (four oxygen atoms) at the maximum can exists in hydroperoxide groups indicated by a signal shift of two nominal mass units. The corresponding product belongs to the "norm. AutOx." group, O,O-$C_{15}H_{23-x}(OOH)_xO_2$ with x = 2, species **11** in Fig. 7. The signal intensity of the signal shifted by two nominal mass units accounts for 31 % of the total signal intensity, see the red peak at nominal 395 Th. On the other

hand, the signal shift by one nominal mass unit less, blue peak at nominal 394 Th, can only be explained by an oxygen molecule insertion without forming a hydroperoxide group. This insertion is tentatively explained by an endoperoxide formation from the internal reaction of a RO₂ radical with the second, still intact, double bond in the molecule, see reaction sequence **5'** → **8**



→ **9** in Fig. 6 and **7'** → **13** → **14** in Fig. 7. The signal intensity of this reaction product from the extended autoxidation mechanism "ext. AutOx.", $O,O\text{-}C_{15}H_{23-y}(OO)(OOH)_yO_2$ with $y = 1$, accounts for 69 % of the total intensity of the shifted peaks. The group "(OO)" in the formula stands for the inserted oxygen molecule appearing as the postulated endoperoxide, see species **14** in Fig. 7.

The signal of the $RO_2$ radical at nominal 425 Th was shifted by three or two nominal mass units accounting for 29 % and 71 % of the total signal intensity, respectively. Here, compared to the reaction products appearing at nominal 393 Th, a next oxygen molecule was inserted in the products resulting in a third hydroperoxide group in "norm. AutOx.", $O,O\text{-}C_{15}H_{23-x}(OOH)_xO_2$ with $x = 3$, and a second hydroperoxide group in "ext. Autox.", $O,O\text{-}C_{15}H_{23-y}(OO)(OOH)_yO_2$ with $y = 1$. The signal of the corresponding closed-shell product to the $RO_2$ radical at nominal 425 Th is visible at nominal 408 Th. It

shows the same signal shift as its corresponding $RO_2$ radical by three or two nominal mass units. The signal intensity of the closed-shell product from the "norm. AutOx." group, $O,O\text{-}C_{15}H_{22-x}O(OOH)_x$ with $x = 3$ accounts for 30 % of the total signal intensity of the shifted peaks (red peak at nominal 411 Th), the signal intensity of the reaction product from the "ext. AutOx. group, $O,O\text{-}C_{15}H_{22-y}O(OO)(OOH)_yO_2$ with $y = 2$ (blue peak at 410 Th) accounts for 70 %.

The relative contributions of the two product groups to the total signal intensity for all signals are summarized in Table 1. With

the exception of the signal at nominal 376 Th, the ratio of the contributions of the two product groups is "norm.AutOx." / "ext. AutOx." = 3/7-2/8. This ratio shows, that the extended autoxidation mechanism is more important than the normal autoxidation mechanism for reaction products from the ozonolysis of β-caryophyllene.

Figure 4 shows a comparison of spectra in the nominal mass-to-charge range 345-385 Th recorded in the presence and absence of $D_2O$. The detected signals at nominal 349, 364 and 381 Th are assigned to the third product group "ext. AutOx -$CO_2$". The

signal at nominal 349 Th was shifted by one nominal mass unit when adding $D_2O$ which indicates the presence of one hydroperoxide moiety in this reaction product. This signal has the molecular formula $C_{14}H_{23}O_6$ and one of the six oxygen atoms arises from the initial ozone attack as observed from the experiments with isotopically labeled ozone, see Sect. 3.2. Two oxygen atoms are assigned to the $RO_2$ radical functional group. The signal shift by one nominal mass unit from the H/D exchange experiment indicates that two of the three remaining oxygen atoms form a hydroperoxide moiety. The third, residual

oxygen atom must be inserted into the molecule without generating an additional acidic H atom, illustrated by "(O)" in the general formula $O\text{-}C_{14}H_{23-\alpha}(O)(OOH)_\alpha O_2$. The chemical nature of this "(O)" in the product is still uncertain and was tentatively attributed to an epoxide formation at the second double bond, see **7'** → **15** in Fig. 7 and the discussion in the Sect. before (3.2). The position of the $RO_2$ radical signal of $O\text{-}C_{14}H_{23-\alpha}(O)(OOH)_\alpha O_2$ with $\alpha = 2$ at nominal 381 Th and its corresponding closed-shell product $C_{14}H_{22}O_7$ at nominal 364 Th were shifted by two nominal mass units in presence of $D_2O$. The insertion of a next

oxygen molecule leads to the formation of the $RO_2$ radical $O\text{-}C_{14}H_{23-\alpha}(O)(OOH)_\alpha O_2$ with $\alpha = 3$ detected at nominal 413 Th, and its closed-shell product at nominal 396 Th. Both signals were shifted by three nominal mass units in presence of $D_2O$. Signals of reaction products from the "ext. AutOx. -$CO_2$" group with more than ten oxygen atoms and more than three hydroperoxide moieties were not detected.



## 3.4 Mechanism of HOM formation

Figures 5-7 show the proposed initial reaction steps of the ozonolysis of β-caryophyllene with a focus on the HOM formation. The reaction is initiated by the ozone attack at the more reactive, endocyclic double bond of β-caryophyllene **1** marked by the orange oval in Fig. 5. The rate coefficient of the reaction of ozone with the endocyclic double bond is about 100 times higher

than that of the exocyclic double bond (Winterhalter et al., 2009). Therefore, the reaction of ozone with the exocyclic double bond is neglected here. The reaction of ozone with a double bond is exothermic and forms carbonyl oxides, the so-called Criegee intermediates (CIs), **2a** and **2b** (Criegee, 1975). Due to the reaction exothermicity, the CIs exist initially with a large amount of excess energy (chemically activated CIs), which is stepwise lost by collisions with the bath gas molecules (Kroll et al., 2001). CIs with an internal energy below a definite threshold energy, needed for prompt decomposition, are called stabilized

CIs (Vereecken and Francisco, 2012). Both, stabilized and chemically activated CIs, can undergo unimolecular reactions or can be further collisionally stabilized by the bath gas (Kroll et al., 2001; Vereecken et al., 2012). Stabilized CIs can also react in a variety of bimolecular reactions depending on their molecular structure (Vereecken et al., 2012). An important unimolecular isomerization step gives the corresponding vinyl hydroperoxide **3a**, **3b** and **3c** (Drozd et al., 2011; Kroll et al., 2001; Vereecken et al., 2012) that further decomposes under OH radical release and formation of the alkyl radicals **4a**, **4b** and

**4c**. For simplicity, the reaction scheme does not differentiate between excited and stabilized molecules.

Figure 6 focuses on further reaction pathways of the alkyl radical **4b**. It is supposed that **4a** and **4c** are reacting similarly. Molecular oxygen rapidly adds to **4b** forming the first RO$_2$ radical **5**. Species **5** can either react via an intramolecular H-transfer, **5 → 6**, followed by O$_2$ addition forming the RO$_2$ radical **7** from the product group "norm. AutOx.", O,O-C$_{15}$H$_{23-x}$(OOH)$_x$O$_2$ with x = 1, or **5** can internally attack the remaining double bond forming an endoperoxide and an alkyl

radical, **5 → 5' → 8,** and after O$_2$ addition the RO$_2$ radical **9**. This cyclisation leads to an O$_2$ insertion without forming a hydroperoxide moiety, indicated by "(OO)" in the formula O,O-C$_{15}$H$_{23-y}$(OO)(OOH)$_y$O$_2$ of the product group "ext. AutOx.". "(OO)" represents the endoperoxide group. The RO$_2$ radical **9** can be further oxidized via the autoxidation mechanism forming RO$_2$ radicals belonging to the product group "ext. AutOx.", O,O-C$_{15}$H$_{23-y}$(OO)(OOH)$_y$O$_2$ with y = 1-4, not shown here. A similar endoperoxide formation was already predicted for the OH radical-initiated oxidation of aromatic compounds (Andino

et al., 1996; Bartolotti and Edney, 1995; Berndt and Böge, 2006; Ghigo and Tonachini, 1999; Suh et al., 2003). Berndt et al. (2015b) validated the formation of endoperoxide-group containing RO$_2$ radicals from the OH radical-initiated oxidation of mesitylene (1,3,5-trimethylbenzene) based on the detection of accretion products of these RO$_2$ radicals. Endoperoxide formation was also proposed from theoretical investigations for the reaction of OH radicals with the monoterpenes α- and β-pinene (Vereecken et al., 2007; Vereecken and Peeters, 2004; Vereecken and Peeters, 2012) and tentatively confirmed in

chamber experiments (Eddingsaas et al., 2012).

Figure 7 shows the further reaction pathways of the RO$_2$ radical **7** from the "norm. AutOx." group, O,O-C$_{15}$H$_{23-x}$(OOH)$_x$O$_2$ with x = 1. The step **7 → 10a → 11** is an intramolecular H-transfer with subsequent O$_2$ addition under formation of the RO$_2$ radical **11**, O,O-C$_{15}$H$_{23-x}$(OOH)$_x$O$_2$ with x = 2 ("norm. AutOx."). Furthermore, the closed-shell product **12** can be formed via





intramolecular H-transfer and subsequent OH radical elimination, $7 \rightarrow 10b \rightarrow 12$. The formation of HOMs from the "ext. AutOx." group can be explained by the internal $RO_2$ radical reaction with the remaining double bond. This might lead to the cyclization product **13** that subsequently adds $O_2$ forming the next $RO_2$ radical **14**, $O,O\text{-}C_{15}H_{23-y}(OO)(OOH)_yO_2$ with y = 1. The formation of HOMs from the product group "ext. AutOx. $-CO_2$" is uncertain at the moment. A possible reaction sequence

starting from the $RO_2$ radical **7** is shown in Fig. 7, $7 \rightarrow 7' \rightarrow 15 \rightarrow 16 \rightarrow 17$. In this reaction mechanism an epoxidation step is proposed, $7' \rightarrow 15$. Subsequently, $CO_2$ is eliminated from the acylalkoxy radical functional group, $15 \rightarrow 16$, resulting in an alkyl radical **16** that rapidly adds $O_2$ forming the $RO_2$ radical **17**. This new $RO_2$ radical **17**, $O\text{-}C_{14}H_{23-\alpha}(O)(OOH)_\alpha O_2$ with $\alpha = 1$, can further react via autoxidation, i.e. intramolecular H-transfer and subsequent $O_2$ addition, forming the next $RO_2$ radicals of the "ext. AutOx. $-CO_2$" group with $\alpha = 2$ and 3.

The epoxide formation cannot be proven and represents only a proposed reaction pathway in order to explain the experimental results. A similar epoxide formation step was postulated for the OH radical-initiated oxidation of aromatic compounds (Bartolotti and Edney, 1995; Glowacki et al., 2009; Motta et al., 2002; Pan and Wang, 2014; Suh et al., 2003; Yu and Jeffries, 1997). Possible reaction products, e.g. epoxide carbonyls, were detected in small quantities using GC-MS analysis (Glowacki et al., 2009; Yu and Jeffries, 1997).

The Figures 5-7 show the proposed reaction paths leading to the first $RO_2$ radicals of all three product groups. Consecutive oxidation processes lead to the next $RO_2$ radicals in competition to termination reactions like $RO_2 + R'O_2$, or $RO_2 + NO$. The formation of first-generation closed-shell products from highly oxidized $RO_2$ radicals is discussed by Jokinen et al. (2014) and is not included here.

### 3.5 Experiment with addition of the sCI scavenger CH₃COOH

A measurement series in the presence of acetic acid ($CH_3COOH$) has been conducted in order to get an indication whether the HOM formation starts from the chemically activated CI or from the collisionally stabilized CI (sCI), species **2a** and **2b** in Fig. 8. Small organic acid were found to efficiently react with sCIs (Beck et al., 2011; Neeb et al., 1996) while chemically activated CIs exclusively react via unimolecular reactions, and bimolecular reactions with other species (such as acids) can be neglected (Vereecken and Francisco, 2012) , see also Sect. 3.4.

Figure 8 shows the concentrations of three highly oxidized $RO_2$ radicals from the three product groups as a function of the acetic acid ($CH_3COOH$) concentration in the reaction gas. The analysis has been done using nitrate ionization. Additionally, also acetic acid was detectable by the $(CH_3COOH)NO_3^-$ adduct. The stated (lower limit) adduct concentrations are by a factor of $2 \times 10^7$ smaller than the acetic acid concentration in the reaction gas. Even for the highest $CH_3COOH$ concentrations of $1.4 \times 10^{14}$ molecules $cm^{-3}$, no influence of the HOM concentrations on the acid concentration was detected, see Fig. 8.

The absolute rate coefficient of the reaction of acetic acid with sCIs ($CH_2OO$ or $CH_3CHOO$) was measured at 4 torr and 298 K to $(1.2\text{-}2.5) \times 10^{-10}$ $cm^3$ molecule$^{-1}$ s$^{-1}$ (Welz et al., 2014). Assuming a value of $2 \times 10^{-10}$ $cm^3$ molecule$^{-1}$ s$^{-1}$ for the rate coefficient of the reaction of acetic acid with the sCIs from β-caryophyllene ozonolysis, a sCI lifetime with respect to this reaction of 3.6



x $10^{-5}$ s at [CH$_3$COOH] = 1.4 x $10^{14}$ molecules cm$^{-3}$ follows. The sCI lifetime with respect to the unimolecular reactions, **2a** → **3a** and **2b** → **3b/c**, is substantial longer with 4 x $10^{-3}$ s assuming the kinetic data for the largest sCI ((CH$_3$)$_2$COO) available (Olzmann et al., 1997). That means that at [CH$_3$COOH] > $10^{13}$ molecules cm$^{-3}$, the fate of the sCIs is most likely dominated by the reaction with CH$_3$COOH and the formation of **3a**-**3c**, the expected precursors species of the HOMs, is suppressed. The absence of any effect of the HOM concentrations on the acetic acid concentration is taken as an indicator, that the sCIs are not involved in the HOM formation. Consequently, the HOM formation is provisionally attributed to reactions starting from the chemically activated Criegee intermediates.

### 3.6 Time dependence of RO$_2$ radical formation

All previous experiments were conducted with a reaction time of 7.9 s. A variation of the reaction time allowed to examine the possible time dependence of the reaction processes.

Therefore, the reaction time was varied for constant initial conditions in the time range of 3.0-7.9 s using acetate ionization and the concentration changes of RO$_2$ radical from all three product groups were investigated, see Fig. 9. All RO$_2$ radical concentrations increased proportionally with time. That shows first, that no significant RO$_2$ radical consumption occurred at these reaction conditions. Secondly, the interconversion of all RO$_2$ radicals, including the RO$_2$ radicals from the "ext. AutOx" group with a proposed endoperoxide formation and those from the "ext. AutOx. -CO$_2$" group with a proposed CO$_2$ elimination, proceeds at a time scale of seconds, i.e. with a rate coefficient $\geq$ 1 s$^{-1}$. The RO$_2$ concentrations increased by a factor of 2.3-2.7 from the shortest to the longest reaction time which is close to the time increase factor of 2.6. This finding differs from the results of an investigation of cyclohexene ozonolysis using the same experimental setup where a concentration increase by a factor of 20-35 was detected when extending the reaction time from 1.5 to 7.9 s (Berndt et al., 2015b). This strong increase was explained by the presence a rate limited entrance channel for the highly oxidized RO$_2$ radicals detected from the cyclohexene ozonolysis. A similar behavior was not observed for the formation of highly oxidized RO$_2$ radicals from β-caryophyllene ozonolysis.

## 4 Conclusion

The mechanism of the formation of highly oxidized multifunctional organic compounds (HOMs) from the ozonolysis of β-caryophyllene was investigated in a free-jet flow system at atmospheric pressure and a temperature of 295 ± 2 K. β-Caryophyllene is globally the most emitted sesquiterpene, responsible for up to 70 % of regional biogenic volatile organic compound emissions, e.g. in orange orchards (Ciccioli et al., 1999). It is mainly oxidized by ozone under atmospheric conditions. The HOM formation from this reaction was recently studied in this laboratory (Richters et al., 2016). Different reaction products were detected, that could not be assigned to the class of highly oxidized RO$_2$ radicals formed via the "normal" autoxidation mechanism (Jokinen et al., 2014; Richters et al., 2016). This behavior was attributed to the presence of a second




double bond in β-caryophyllene which enables further reaction channels. These, up to now undiscovered reaction pathways were investigated with the help of labeling experiments using heavy water and isotopically labeled ozone ($^{18}O_3$). The experimental results allowed to tentatively postulate extended autoxidation mechanisms including i) the formation of an endoperoxide moiety in the $RO_2$ radicals ("ext. AutOx" group) and ii) a $CO_2$ elimination in presence of an unsaturated peroxy

acyl radical ("ext. AutOx -$CO_2$" group).

Time-dependent investigations of the formation of highly oxidized $RO_2$ radicals showed that all $RO_2$ radicals are formed on a time scale of less than three seconds. Experiments with acetic acid, serving as a scavenger of stabilized Criegee intermediates, indicated that HOM formation most likely proceeds via reactions of the chemically excited Criegee intermediates formed as an early reaction product from the ozonolysis of β-caryophyllene.

In conclusion, this study provides insights in new reaction pathways that extend the autoxidation mechanism for unsaturated $RO_2$ radicals in the gas phase. About two thirds of the total molar HOM yield from the ozonolysis of β-caryophyllene can be explained with the help of these new reaction pathways. Further work is needed to validate the proposed reaction steps of the extended autoxidation mechanism.

**Acknowledgement**

We would like to thank K. Pielok and A. Rohmer for technical assistance and the German Academic Scholarship Foundation (Studienstiftung des deutschen Volkes) for funding.



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





**Table 1.** Highly oxidized reaction products from the ozonolysis of β-caryophyllene detected as nitrate ion adducts and acetate ion adducts using CI-APi-TOF mass spectrometry. Products were categorized into three product groups, i.e. "norm. AutOx.", "ext. AutOx." and "ext. AutOx. -CO$_2$". Signals from the "norm. AutOx." and "ext. AutOx." groups were detected at the same mass-to-charge ratio. The percentages indicate the contribution of a signal to the different product groups, "norm. AutOx." and "ext. AutOx.", as elucidated by H/D exchange experiments using nitrate ionization.

| Nominal mass-to-charge ratio | | Molecular formula | Product group (contribution to the total signal (%)) | | RO$_2$ radical | Closed-shell product |
|---|---|---|---|---|---|---|
| nitrate ion adducts | acetate ion adducts | | | | | |
| 349 | 346 | C$_{14}$H$_{23}$O$_6$ | ext. AutOx. -CO$_2$ | | O-C$_{14}$H$_{22}$(O)(OOH)O$_2$ | |
| 361 | 358 | C$_{15}$H$_{23}$O$_6$ | norm. AutOx. | | O,O-C$_{15}$H$_{22}$(OOH)O$_2$ | |
| 364 | 361 | C$_{14}$H$_{22}$O$_7$ | ext. AutOx. -CO$_2$ | | | O-C$_{14}$H$_{20}$O(O)(OOH)$_2$ |
| 376 | 373 | C$_{15}$H$_{22}$O$_7$ | norm. AutOx. | (56 %) | | O,O-C$_{15}$H$_{20}$O(OOH)$_2$ |
| | | C$_{15}$H$_{22}$O$_7$ | ext. AutOx. | (44 %) | | O,O-C$_{15}$H$_{21}$O(OO)(OOH) |
| 381 | 378 | C$_{14}$H$_{23}$O$_8$ | ext. AutOx. -CO$_2$ | | O-C$_{14}$H$_{21}$(O)(OOH)$_2$O$_2$ | |
| 393 | 390 | C$_{15}$H$_{23}$O$_8$ | norm. AutOx. | (31 %) | O,O-C$_{15}$H$_{21}$(OOH)$_2$O$_2$ | |
| | | | ext. AutOx. | (69 %) | O,O-C$_{15}$H$_{22}$(OO)(OOH)O$_2$ | |
| 396 | 393 | C$_{14}$H$_{22}$O$_9$ | ext. AutOx. -CO$_2$ | | | O-C$_{14}$H$_{19}$O(O)(OOH)$_3$ |
| 408 | 405 | C$_{15}$H$_{22}$O$_9$ | norm. AutOx. | (30 %) | | O,O-C$_{15}$H$_{19}$O(OOH)$_3$ |
| | | C$_{15}$H$_{22}$O$_9$ | ext. AutOx. | (70 %) | | O,O-C$_{15}$H$_{20}$O(OO)(OOH)$_2$ |
| 413 | 410 | C$_{14}$H$_{23}$O$_{10}$ | ext. AutOx. -CO$_2$ | | O-C$_{14}$H$_{20}$(O)(OOH)$_3$O$_2$ | |
| 425 | 422 | C$_{15}$H$_{23}$O$_{10}$ | norm. AutOx. | (29 %) | O,O-C$_{15}$H$_{20}$(OOH)$_3$O$_2$ | |
| | | C$_{15}$H$_{22}$O$_9$ | ext. AutOx. | (71 %) | O,O-C$_{15}$H$_{21}$(OO)(OOH)$_2$O$_2$ | |
| 440 | 437 | C$_{15}$H$_{22}$O$_{11}$ | norm. AutOx. | (29 %) | | O,O-C$_{15}$H$_{18}$O(OOH)$_4$ |
| | | C$_{15}$H$_{22}$O$_{11}$ | ext. AutOx. | (71 %) | | O,O-C$_{15}$H$_{19}$O(OO)(OOH)$_3$ |
| 457 | 454 | C$_{15}$H$_{23}$O$_{12}$ | norm. AutOx. | (25 %) | O,O-C$_{15}$H$_{19}$(OOH)$_4$O$_2$ | |
| | | C$_{15}$H$_{23}$O$_{12}$ | ext. AutOx. | (75 %) | O,O-C$_{15}$H$_{20}$(OO)(OOH)$_3$O$_2$ | |
| 472 | 469 | C$_{15}$H$_{22}$O$_{13}$ | norm. AutOx. | (22 %) | | O,O-C$_{15}$H$_{17}$O(OOH)$_5$ |
| | | C$_{15}$H$_{22}$O$_{13}$ | ext. AutOx. | (78 %) | | O,O-C$_{15}$H$_{18}$O(OO)(OOH)$_4$ |
| 489 | 486 | C$_{15}$H$_{23}$O$_{14}$ | norm. AutOx. | (22 %) | O,O-C$_{15}$H$_{18}$(OOH)$_5$O$_2$ | |
| | | C$_{15}$H$_{23}$O$_{14}$ | ext. AutOx. | (78 %) | O,O-C$_{15}$H$_{19}$(OO)(OOH)$_4$O$_2$ | |



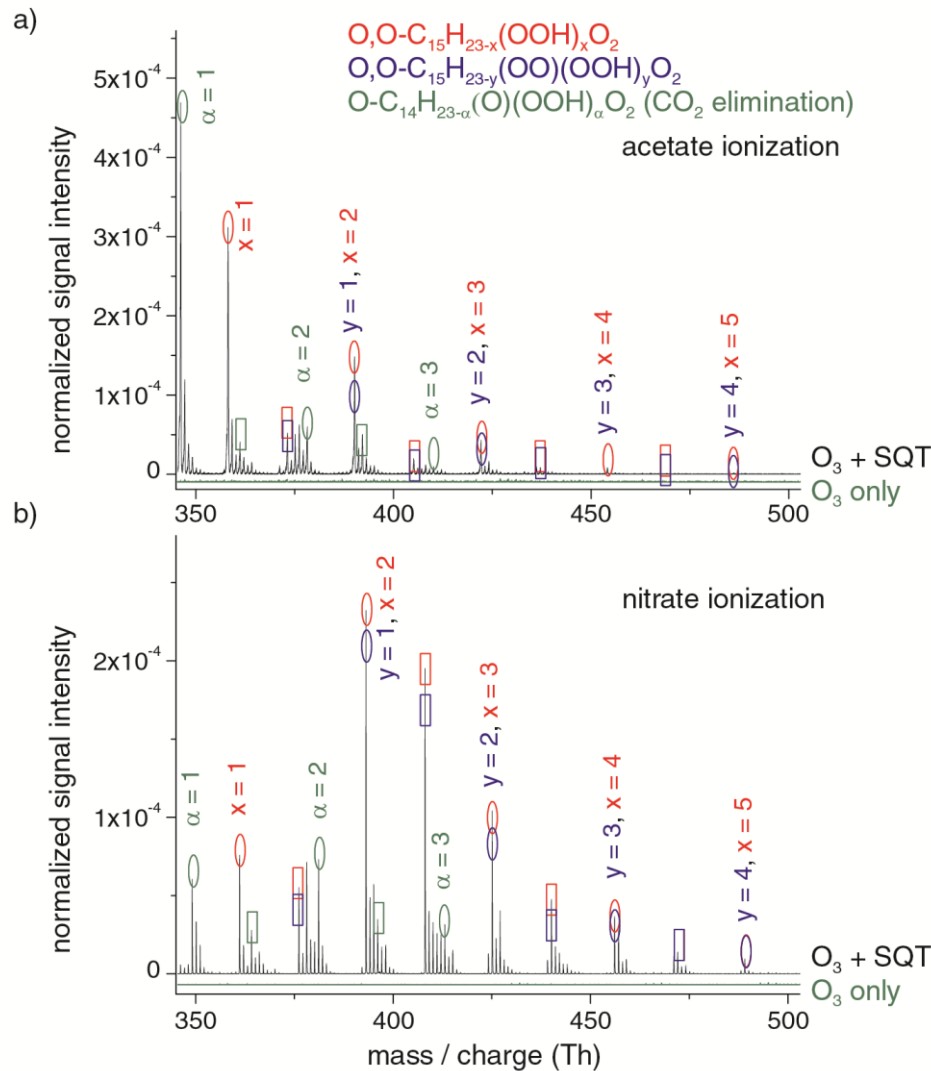

**Figure 1.** Highly oxidized $RO_2$ radicals of the three product groups "norm. AutOx.", $O,O\text{-}C_{15}H_{23-x}(OOH)_xO_2$ with x = 1-5 (in red), "ext. AutOx.", $O,O\text{-}C_{15}H_{23-y}(OO)(OOH)_yO_2$ with y = 1-4 (in blue) and "ext. AutOx -$CO_2$", $O\text{-}C_{14}H_{23-\alpha}(O)(OOH)_\alpha O_2$ with $\alpha$ = 1-3 (in green), and corresponding closed-shell products (rectangular lines) appearing at -17 nominal mass units regarding the corresponding $RO_2$ radical. The products were detected by means of a) acetate ionization and b) nitrate ionization. The same molecule gives a signal shifted by three nominal mass units comparing the acetate ion adducts (+59 nominal mass units) with the nitrate ion adducts (+62 nominal mass units). The mass spectra were normalized by their reagent ion counts. Signals from the "norm. AutOx." group and the "ext. AutOx." group were detected at the same mass-to-charge ratio. The green spectrum lines ($O_3$ only) shows the background experiments in which only ozone (no SQT) was present. The data of Fig. 1b was taken from Richters et al. (2016). [β-caryophyllene] = 8.6 x $10^{10}$ (acetate ionization); [β-caryophyllene] = 8.3 x $10^{10}$ (nitrate ionization); [$O_3$] = 1.02 x $10^{12}$ molecules cm$^{-3}$; reaction time: 7.9 s.





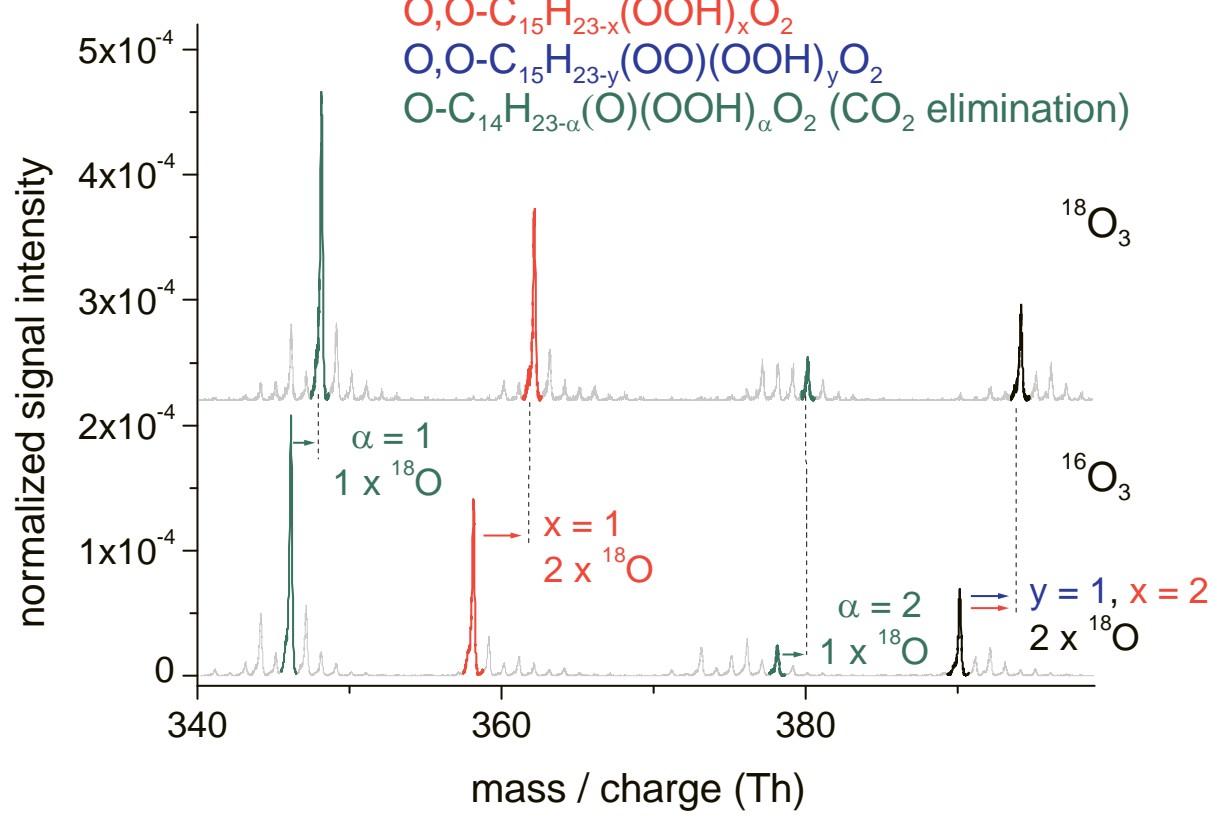

**Figure 2.** Ozonolysis of β-caryophyllene using $^{16}O_3$ (lower part) and $^{18}O_3$ (upper part) and applying acetate ionization in the analysis. Highly oxidized $RO_2$ radicals of the three product groups "norm. AutOx.", $O,O-C_{15}H_{23-x}(OOH)_xO_2$ with x = 1 and 2 (in red), "ext. AutOx.", $O,O-C_{15}H_{23-y}(OO)(OOH)_yO_2$ with y = 1 (in blue) and "ext. AutOx -$CO_2$", $O-C_{14}H_{23-\alpha}(O)(OOH)_\alpha O_2$ with α = 1 and 2 (in green) were detected. The black colored signals at nominal 390 Th ($^{16}O_3$) and nominal 394 Th ($^{18}O_3$) stand for the sum of the signal from the "norm. AutOx." $RO_2$ radical $O,O-C_{15}H_{23-x}(OOH)_xO_2$ with x = 2 and from the "ext. AutOx." $RO_2$ radical $O,O-C_{15}H_{23-y}(OO)(OOH)_yO_2$ with y = 1. Only the arrows and inscriptions (y = 1; x = 2) indicate the colors of the product groups. When exchanging $^{16}O_3$ by $^{18}O_3$, the signals were shifted by two nominal mass units for each oxygen atom arising from the initial ozone attack. [β-caryophyllene] = 8.6 x $10^{10}$; [$O_3$] = 8.8 x $10^{11}$ molecules cm$^{-3}$; reaction time: 7.9 s.





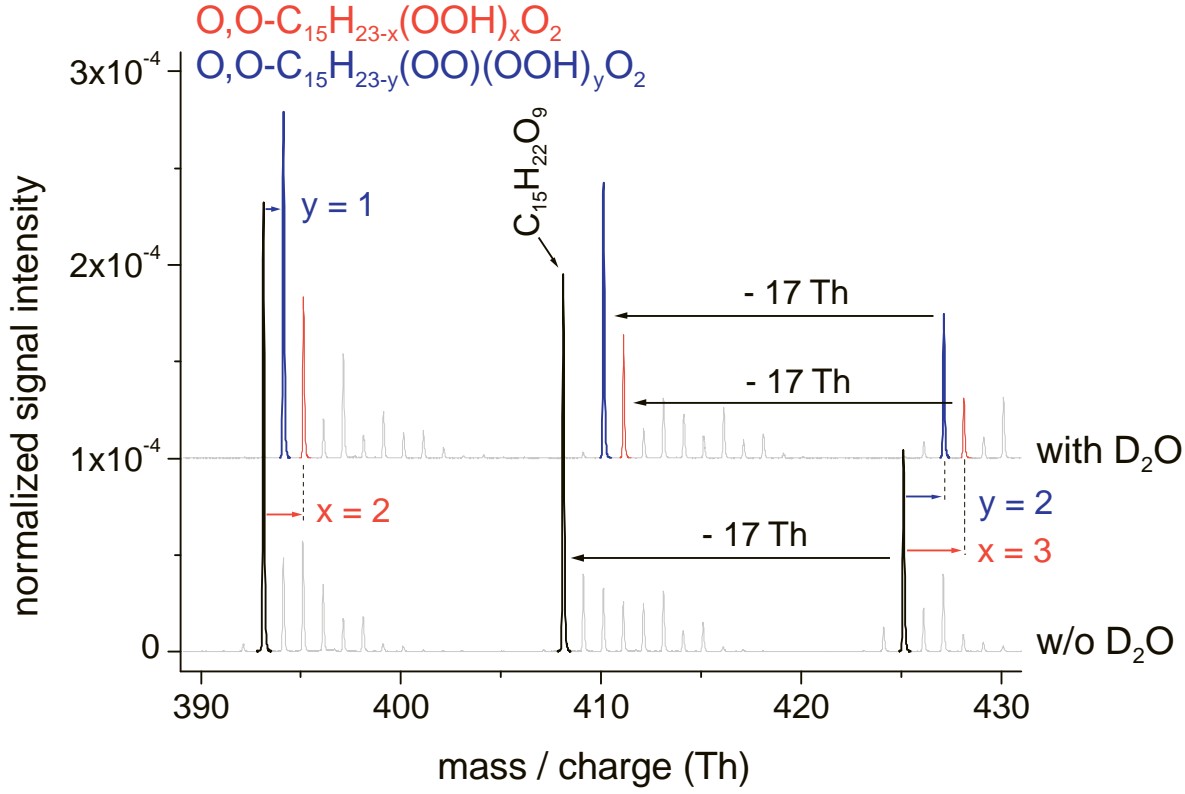

**Figure 3.** Ozonolysis of β-caryophyllene, in absence (lower part) and presence (upper part) of $D_2O$ applying nitrate ionization in the analysis. Signals highlighted in black stand for the sum of signals in absence of $D_2O$ from highly oxidized $RO_2$ radicals of the product groups "norm. AutOx.", $O,O\text{-}C_{15}H_{23-x}(OOH)_xO_2$ with x = 2 and 3, and "ext. AutOx.", $O,O\text{-}C_{15}H_{23-y}(OO)(OOH)_yO_2$, with y = 1 and 2, and the corresponding closed-shell product $C_{15}H_{22}O_9$ of the $RO_2$ radicals for x = 3 or y = 2. The addition of $D_2O$ leads to an H/D exchange of all acidic H atoms present in the molecule. With that, signals from the two product groups are separated by their number of acidic H atoms and the split-up signals are highlighted in red for the "norm. AutOx." group and in blue for the "ext. AutOx." group. [β-caryophyllene] = 8.3 x $10^{10}$; $[O_3]$ = 1.02 x $10^{12}$ molecules $cm^{-3}$; reaction time: 7.9 s.





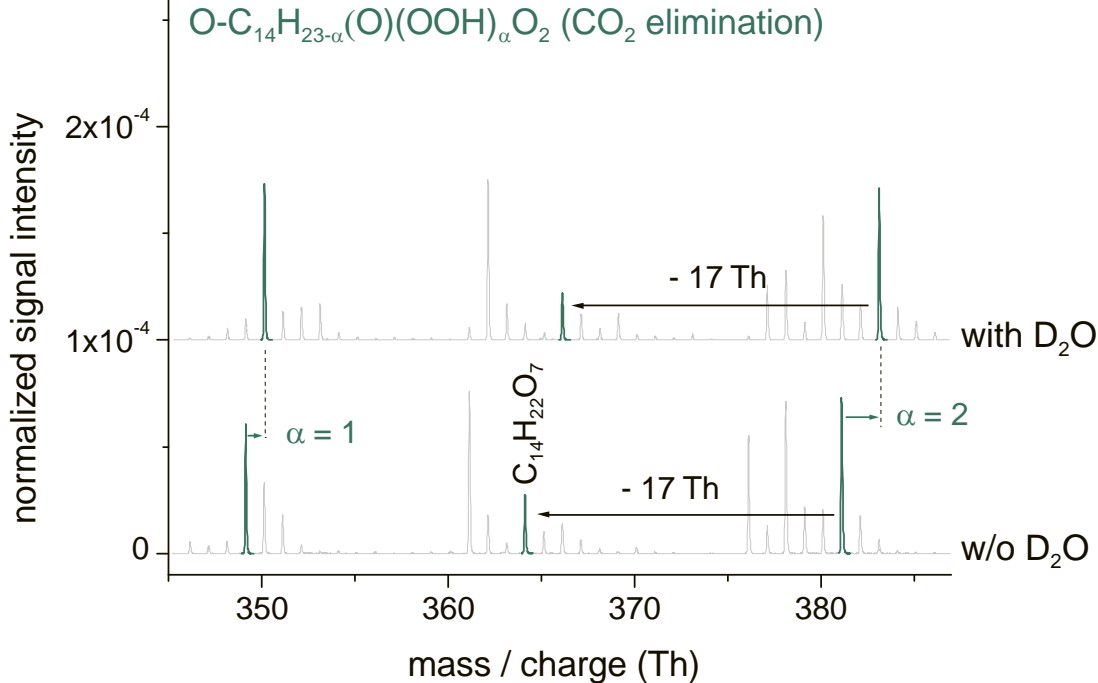

**Figure 4.** Ozonolysis of β-caryophyllene in absence (lower part) and presence (upper part) of $D_2O$ applying nitrate ionization in the analysis. Highly oxidized $RO_2$ radicals of the product group "ext. AutOx. $-CO_2$", $O-C_{14}H_{23-\alpha}(O)(OOH)_\alpha O_2$ with $\alpha = 1$ and 2 and the corresponding closed-shell product ($C_{14}H_{22}O_7$) of the $RO_2$ radical with $\alpha = 2$ are highlighted in green. The addition of $D_2O$ leads to an H/D exchange of the acidic H atoms being equal to the number of hydroperoxide groups in the molecules, i.e. a shift by one nominal mass unit for $\alpha = 1$ or a shift by two nominal mass units for $\alpha = 2$ (including the corresponding closed-shell product). [β-caryophyllene] = 8.3 x $10^{10}$; $[O_3] = 1.02$ x $10^{12}$ molecules $cm^{-3}$; reaction time: 7.9 s.





**Figure 5.** First reaction steps of the ozonolysis of β-caryophyllene. The attack of the more reactive endocyclic double bond (highlighted in orange) is exclusively demonstrated. Oxygen atoms arising from the attacking ozone are highlighted in blue, the alkyl radical functional groups with a shaded oval.



**Figure 6.** Further reaction steps of the alkyl radical 4b. Oxygen atoms arising from the attacking ozone are highlighted in blue, alkyl radical functional groups with a shaded oval and RO₂ radical functional groups with a shaded rectangle. Detected species are surrounded by a solid rectangle. The stated position, where the internal H-transfer takes place 5 → 6, represents an examples only.







**Figure 7.** Further reaction steps of the RO₂ radical (7). Oxygen atoms arising from the attacking ozone are highlighted in blue, alkyl radical functions with a shaded oval and RO₂ radical functional groups with a shaded rectangle. Detected species are surrounded by a solid rectangle. The stated position, where the internal H-transfer takes place 7 → 10a, represent an example only. The dashed arrows indicate that the stated

5    reaction pathway remains uncertain.





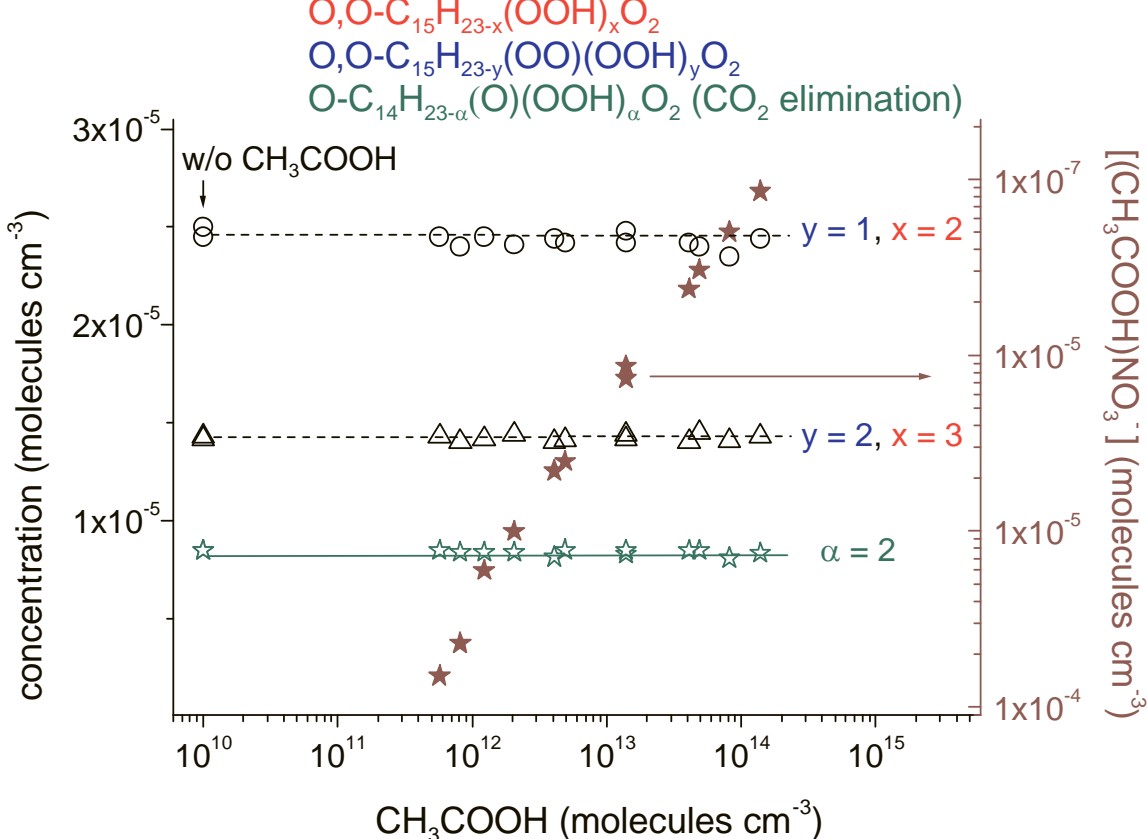

**Figure 8:** Highly oxidized $RO_2$ radicals from the ozonolysis of β-caryophyllene from the three product groups "norm. AutOx." with O,O-$C_{15}H_{23-x}(OOH)_xO_2$, x = 2 and 3, "ext. AutOx." with O,O-$C_{15}H_{23-y}(OO)(OOH)_yO_2$, y = 1 and 2, and "ext. AutOx. -$CO_2$" with O-$C_{14}H_{23-\alpha}(O)(OOH)_\alpha O_2$, α = 2 as a function of the $CH_3COOH$ concentration. $CH_3COOH$ acts as a sCI scavenger. The black colored data points stand for the $RO_2$ radicals from the "norm. AutOx" group and from the "ext. AutOx" group with x = 2 and y = 1 (circle) as well as with x = 3 and y = 2 (triangle). The adduct $(CH_3COOH)NO_3^-$ was detected with lower-limit concentrations which are a factor of $2 \times 10^7$ lower than the acetic acid concentration in the tube. [β-caryophyllene] = $8.3 \times 10^{10}$; [$O_3$] = $4.7 \times 10^{10}$; [$CH_3COOH$] = (0-1.4) $\times 10^{14}$ molecules cm⁻³; reaction time: 7.9 s.



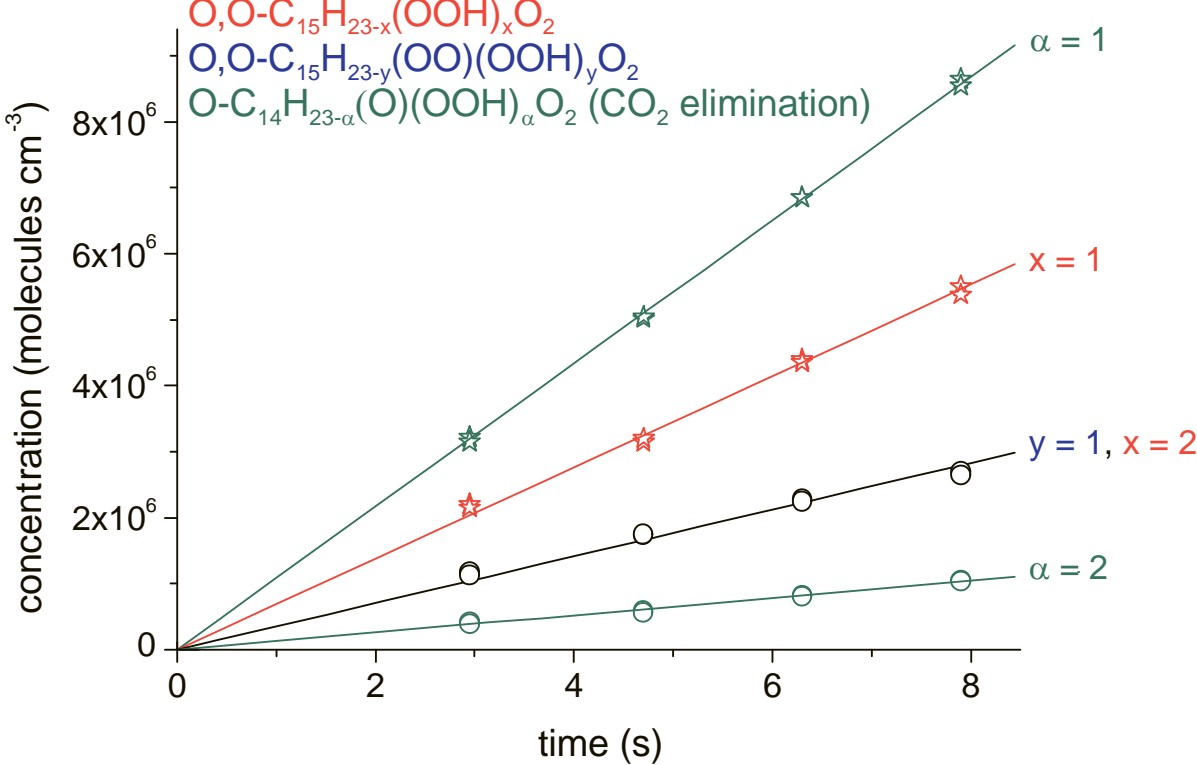

**Figure 9.** Time dependence of highly oxidized $RO_2$ radical formation from the ozonolysis of β-caryophyllene using acetate ionization, data from the "norm. AutOx." group with O,O-$C_{15}H_{23-x}(OOH)_xO_2$ with $x = 1$ and 2 (in red), from "ext. AutOx." with O,O-$C_{15}H_{23-y}(OO)(OOH)_yO_2$ with $y = 1$ (in blue) and from "ext. AutOx. -$CO_2$" with O-$C_{14}H_{23-\alpha}(O)(OOH)_\alpha O_2$ with $\alpha = 1$ and 2 (in green). The black colored data points (open circles) stand for the sum of the $RO_2$ radical from the "norm. AutOx" group, O,O-$C_{15}H_{23-x}(OOH)_xO_2$ with $x = 2$ and from the "ext. AutOx" group, O,O-$C_{15}H_{23-y}(OO)(OOH)_yO_2$ with $y = 1$. [β-caryophyllene] = $8.6 \times 10^{10}$; [$O_3$] = $3.1 \times 10^{11}$ molecules $cm^{-3}$; reaction time: 3.0-7.9 s.