# Peer review of "Different Pathways of the Formation of Highly Oxidized Multifunctional Organic Compounds (HOMs) from the Gas-Phase Ozonolysis of $\beta$ -Caryophyllene"

_Atmospheric Chemistry and Physics, 2016_

## Referee Comment (RC1) · Anonymous Referee #1 · 13 Apr 2016

The authors describe a product study of caryophyllene ozonolysis in a free jet flow tube. Experiments were performed applying acetate CI-API-TOF-MS and nitrate CI-API-TOF-MS to detect peroxy radicals and closed shell oxidation products. Labelling experiments using heavy $O_3$ and $D_2O$ helped to discriminate product classes from different reaction pathways and were used to underpin the proposed reaction scheme. This is an excellent, original study which was carefully conducted and evaluated. The proposed reaction schemes are reasonable and - wherever possible - supported with earlier findings by the authors and in the literature, although it has some speculative moments. But I see the latter as a challenge and I am wondering if the authors have

ideas how a proof for the scheme in Fig 7 could look like. The paper is well written and very good to read (with a few exceptions, where formulations seem to be a little bit intricate). The results are original and new and give further deep insights into autoxidation and formation of highly oxidized molecules. This excellent manuscript should be published in ACP as it is.

The authors may want to consider the following minor suggestions:

Figures are not addressed in sequence of their numberings in manuscript.

page 5, line 28: "Up to now" does not seem the right intro for what is following.

page 6, line 21: Maybe it is better use "analysis" instead of "investigation" in this context.

page 6, line 26: I think it is better to talk about "HOM signal" instead of "HOM yield", the yield should be the same, independent of the detection scheme.

page 7, line 17: It may be simpler to replace "comparing by using" by "applying".

page 7, line 19: typo, . . . second oxygen atom from the initial ozone attack must "have been" abstracted . . .

page 7, line 28: an acylalkoxy radical "(species 15)". Addition of species number would be helpful.

page 11, line 26: "The analysis has been done using nitrate ionization." may be better "These measurements were performed applying nitrate ionization" ?

page 12, line 10: typo, "of" missing.

Figure 2, caption: Although it is explained later in the caption it confused me that there was no blue signal shown in the Figure. May be better "(in black with blue label)" or so.

Supplement line 15: "deflected" seems to be more appropriate to me than "sucked".

---

## Referee Comment (RC2) · Anonymous Referee #2 · 17 Apr 2016

This is an interesting manuscript describing a continuation of the work from this laboratory on studying autoxidation reactions of biogenic terpenoids. The experiments seem carefully conducted with a previously described instrumentation that is especially suited for studying end product distributions of complex VOC oxidation reactions. However, number of conclusions derived in this paper, especially concerning "the extended autoxidation mechanism", seem somewhat hastily derived - or already reported. Thereby I'm not sure if the manuscript, as it currently stands, brings enough new insight to merit it's publication in ACP.

Most importantly, I do not see a need for "an extended autoxidation mechanism" for various reasons. Firstly, it is of obvious relevance what is understood as "the old mechanism". If the "old mechanism" is only thought of including peroxy radical isomerization + O2 addition steps, then I guess we could talk about a "new mechanism" at some level. However, the whole process is dependent on easily abstractable H-atoms and suitable molecular geometries enabling the abstractions – this is about what is clear at the moment. At the current stage it's not unambiguously clear what kind of steps are needed to progress the autoxidation chain to reach into the most highly-oxidized products in monoterpene oxidation (recently Kurten et al. 2015 suggested that bimolecular steps might be needed to advance the $\alpha$-pinene oxidation). What seems intuitively clear, however, is that we're only beginning to understand the importance and the details of the autoxidation progression. So at this level it seems very preliminary to talk about "extended autoxidation" as we do not have a clear concise picture what constitutes the "normal autoxidation" in this context. In any case, probably you cannot really isolate the different pathways, but can account for branching between pathways under different conditions. So in my humble opinion, there is no need to bring up a "new extended autoxidation mechanism" – it is all the same autoxidation, just with a few additional steps.

Secondly, the pathways suggested to represent this "extended mechanism" constitute unambiguous reaction steps – the CO2 elimination and endoperoxide formation. Importantly, it does not seem to be possible to separate this CO2 elimination pathway from a CO-loss pathway brought up previously in autoxidation studies (e.g., Rissanen 2014, 2015, Mentel 2015). Both of these processes occur from acyl type radicals – the CO loss before O2 addition and the CO2 loss after the O2 addition. So while it is definitely worth to (and you should) discuss the potential of this type of reaction pathway, with current results it is impossible to be sure which type of dissociation process actually occurred. This should be made absolutely clear, and reference to papers where CO-loss was discussed should be given. Thirdly, the endoperoxide formation that is given as the explanation to account for the formation of "too less acidic hydrogens" (see D2O

experiments) has already been suggested in exactly in the same context in previous literature concerning HOM formation (Rissanen et al. 2015, Kurten et al. 2015). So to sum up, it is hard to see the novelty of this paper especially as the main results of $\beta$-caryophyllene ozonolysis were already published previously (see Richters, et al. 2016). Hence, even though the manuscript appears to be generally well made, I cannot support its publication without severe changes in the interpretation of the mechanistic pathways and corresponding modifications for the manuscript text.

Minor points: Some of the terminology seem a bit awkward. In certain places it seems useful to label where the oxygens come from (i.e., O,O-formalism), but in most cases I think it only hinders the reading. So I would propose to stick with common $CxHyOz$ formalism most of the time and then use the more difficult format where you are talking about the mechanism.

Can there be a different detection sensitivity for the peroxy radicals in comparison with the closed-shell products? It seems somewhat counterintuitive that the RO2 radicals could have such a long lifetimes under the present experimental conditions.

How was O3 handled? I assume that the "18O3" flow still contains about 95 to 99% of 18O2 (due to O3 generator generating efficiency) and thereby this could lead to significant difficulties in tracking the amount of O-atoms that are left from ozonolysis and do not result from secondary reactions after the initiation. This would be especially severe in trying to understand the contribution of different pathways (see Section 3.2.).

Page 3, Line 27: The use of CH3COOH is not mentioned. Also in Page 4 and line 13.

Page 4, Line 3: How was caryophyllene sampling done? Did the GC and PTR methods indicate any differences in determined concentrations?

Page 4, line 20: What is meant by mass spectrometer setting?

Page 5, Line 8. The number quoted is only the estimated concentration of highly oxidized RO2. How large do you assume is the pool of other radical species (e.g., how

much are there less oxidized RO2s and HO2)?

Page 5, Lines 20-32: Similar H/D exchange behavior was observed and discussed in Rissanen, et al. 2015.

Page 6, Line 25-25: How can the yield change when changing ionization method? What you want to say is that the detection sensitivity varies between products and ionization methods. But what this means to the determined yields then? Does these yields then mean anything?

Page 7 (and others): Be careful with Figure and Table numbering. Currently there are "Figs." and "Figures" which are not in numerical order.

Page 7, Line 19: Couldn't this be as well accounted for by a reaction in which 18OH (derived from VHP decomposition) starts the autoxidation sequence?

Page 9, Line 10-15: How certain are you that the H/D exchange was 100% complete? For example, in Rissanen et al. 2015 and incomplete H/D shift as seen in reagent ions was shown to result in partial H/D shifts in products too. So how accurate is the determination of the importance of different pathways, based on H/D shift?

Page 10: Would make sense to change section 3.4. to 3.1. to improve the readability.

Page 11, Line 11: Similar epoxide formation is well-known from atmospheric isoprene oxidation (e.g. Paulot et al. 2009)

Page 11, Line 16: RO2 + RO2 is usually considered as progression, not termination.

Page 11, Line 27: Is this the first time nitrate ionization has been reported to see simple carboxylic acids?

Conclusions first sentence: Rather "end-product analysis was used to infer oxidation pathways".

Page 12, Line 26: I think the discussion on VOC sources etc. should be moved to

discussion, after all it's not what was studied in this work.

Page 13, Line 1: "These, up to now undiscovered reaction pathways..." This sentence is an overstatement and simply not true. (In Line 10 a more appropriate wording is used).

Figure 7: I find it a bit odd that at the same time it's stated that the abstraction is "an example only" and then resulting species are said to be detected in the spectra. And the assumed epoxide structure seems questionable.

References: T. Kurtén, et al., J. Phys. Chem. A, 2015, 119,11366. T. Mentel, et al. Atmos. Chem. Phys. 2015, 15, 2791. F. Paulot, et al. 2009, Science, 325, 730. S. Richters, et al. Environ. Sci. Technol. 2016, 50, 2354. M. Rissanen, et al. , J. Am. Chem. Soc. 2014, 136, 15596. M. Rissanen, et al., J. Phys. Chem. A, 2015, 119, 4633.
* * *

---

## Author Comment (AC1) · 21 Apr 2016

Comment to RC2:

We thank the reviewer for the comments that we will carefully take into account for the revised version of our manuscript. In this short reply, we react to the main criticism that the reviewer "is not sure if the manuscript, as it currently stands, brings enough new insight to merit it's publication in ACP" in advance. We agree with the referee, that the "normal" autoxidation mechanism is not completely understood yet. Nevertheless, autoxidation mechanisms were proposed in order to explain the detected product formation from the ozonolysis of cyclohexene (Berndt et al., 2015; Rissanen et al., 2014) and limonene (Jokinen et al., 2014). These reaction schemes can be regarded as the "normal" autoxidation mechanisms in line with the well-known process in the liquid phase. Based on these publications, we believe that the two proposed, extended reaction pathways, "ext. AutOx." and "ext. AutOx. -CO2" in our manuscript, represent indeed real extensions of the "normal" autoxidation mechanism that were not yet shown in the literature. We are aware of the publication by Rissanen et al. (2015) in which it is stated, that additional reaction steps are needed to explain the product formation and the results of H/D exchange experiments from the ozonolysis of alpha-pinene. Indeed, Rissanen et al. (2015) proposed among others also endoperoxide formation based on the literature data from the OH-radical initiated oxidation of aromatic compounds and pinenes. However, alpha-pinene does not contain a second double bond needed for endoperoxide formation as shown for the aromatics. Therefore, an opening of the four-membered ring in the course of the alpha-pinene oxidation was discussed. However, Kurtén et al. (2015) calculated, that the ring opening pathways which include the formation of a double bond, are not likely to occur. We show in our manuscript that the second double bond is indeed crucial for endoperoxide formation allowing to explain the results of the H/D exchange experiments. The structurally similar sesquiterpenes beta-caryophyllene and alpha-cedrene differ mainly by their number of double bonds, two in alpha-caryophyllene and only one in alpha-cedrene. Product spectra and H/D exchange experiments from alpha-cedrene agree perfectly with the "normal" autoxidation mechanism, whereas experimental results for beta-caryophyllene differ and need further reaction pathways for explanation. We think that we can show in our manuscript, that new unimolecular reaction pathways of RO2 radicals in the gas phase are enabled by the presence of a double bond. Certainly, these reaction pathways need further investigations. Overall, we believe that these new reaction pathways for unsaturated RO2 radicals represent an extension of the "normal" autoxidation mechanism in the gas phase not clearly stated before in the literature.

References:

[Figure]

Berndt, T., et al.: (2015), J. Phys. Chem. A 119, 10336-10348. Jokinen, T., et al.: (2014), Angew. Chem., Int. Ed. 53, 14596-14600. Kurtén, T., et al.: (2015), J. Phys. Chem. A 119, 11366-11375. Rissanen, M. P., et al.: (2014), J. Am. Chem. Soc. 136, 15596-15606. Rissanen, M. P., et al.: (2015), J. Phys. Chem. A 119, 4633-4650.

―――――――――――――――――――――――

---

## Referee Comment (RC3) · Anonymous Referee #3 · 2 Jun 2016

The manuscript, "Different Pathways..." by Richters et al. is well written, presents thorough experimental work and detailed spectral analyses that allowed the authors to identify three different pathways by which the Criegee intermediates formed by the ozonolysis of beta-caryophyllene undergo isomerization to become HOMs. The authors utilized a CI-API-ToF-MS with acetate as well as nitrate ionization coupled to a flow-tube. The use of isotopically labeled ozone and water vapor allowed robust identification of the ways in which the various RO2 peroxy radicals are formed, mainly, that a diene (presumably larger than a certain size) can undergo auto-oxidation in a few different ways to form unique molecular products. The branching ratios of these three

pathways are also quantified. The work presented will be a significant contribution to the growing body of research on these HOM species, thus, should be published in ACP with some clarifications.

The D2O experiment (highlighted by figure 3 and SI figure 1) is key to differentiating "norm. AutOx." and "ext. AutOx." since the byproducts of each channel normally (in the absence of D2O) have the same molecular composition. The shift in mass by 1 amu between the y=1 and x=2 products (similarly, y = 2 and x=3) in the D2O experiment allows their distinction. How efficient or fast is the H/D exchange for these acidic H/D atoms? Is it possible that the y=1 peak is really "norm AutOx." product but with just one of the -OOH groups that has undergone H/D exchange?

It appears that (in figures S1 and figure 3) that the red and blue peaks (y=1 and x=2) add up more-or-less to the corresponding black peak (no D2O). Would you expect this to be the case given that a compound with an endoperoxide group should have a different sensitivity (i.e. possess different ion cluster stability) compared to a compound with just hydroperoxide groups? The fact that red and blue add to up black would then suggest y=1 is really x=2 with one less H/D exchange.

Given a reaction time of 7.9 seconds would you expect H/D exchange to be complete? Was there a time dependence experiment conducted (as in section 3.6) with D2O that demonstrates that the peak height for y=1 relative to x=2 does not change with residence or reaction time? If not, perhaps state in the manuscript that a peak where y=1 would reside was not observed for alpha-cedrene (monoalkene) in Richters et al [2016 ES&T figure 3]. Was [D2O] » [H2O] such that at equilibrium essentially all -OOH groups would be present as -OOD?

As reported in lines 15-20 on page 6, the sensitivity (or ion cluster stability) difference between acetate and nitrate ionization to these HOMs (particularly those with one -OOH moiety) is quite large. It is reported (lines 21-30 page 6) that the "norm. AutOx" accounts for "between 29 and 35%" of HOM RO2 formation. This assertion is less

than convincing. The two values do not really represent a range. It is two numbers from two different ionization schemes, each of which detects these compounds with varying efficiencies. There is a factor of 2-3 difference in these branching ratios for the "ext. AutoOx" and "ext. AutoOx -CO2" pathways for the two ionization schemes. Given such large discrepancy, how is reliable or informative are the 29 and 35% numbers or any of these numbers? A discussion is needed on how the varying sensitivities of the ionization schemes are reconciled for this to be quantitative in any way. Moreover, though quantifying an absolute HOM yield is not the objective of this work, I find it uncomfortable that a single sensitivity value obtained from an inorganic acid (H2SO4) is applied to all of these multifunctional organic hydroperoxides and endoperoxides. Though this approach now has become routine, I strongly urge the authors to consider a more robust technique in the future to account for the varying sensitivities (depending on size, ring number, functional group, etc.) to these organics.

Are the authors able to rule out pathways other than the three reported here? Are all peaks in the spectra accounted for by the three pathways? If not, what fraction of the observed peaks (at least the ones that can be reasonably identified as 1st generation products) are attributed to the three pathways? Is the alkoxy radical (RO dot) formation and subsequent degradation/isomerization relevant at all here?

Please include a brief discussion on variables that may affect the branching ratios of the three pathways. Ambient pressure/temperature? Carbon number? The number of rings? Would MT or isoprene undergo "ext. AutoOx" and "ext. AutoOx. -CO2?

In the future, the authors may want to consider not using the blue/red color combination since many have trouble distinguishing the two.

The term "acidic H atoms" is a bit vague. Please re-word in way that doesn't imply that these are H atoms from acid functional groups only. Perhaps "non-alkyl H atoms"?

Line 5 of page 9: "...three or two..." to "...three and two..."

SI figure is really informative and deserves to be in manuscript not SI. Possible to combine with figure 3 or replace figure 3?

---

## Author Response (AR1)

We thank the reviewer for the helpful remarks on our manuscript. Please find our point-by-point responses below.

**Reviewers' comments to author:**

**Reviewer: 1**

Comments:
The authors describe a product study of caryophyllene ozonolysis in a free jet flow tube. Experiments were performed applying acetate CI-API-TOF-MS and nitrate CIAPI-TOF-MS to detect peroxy radicals and closed shell oxidation products. Labelling experiments using heavy O3 and D2O helped to discriminate product classes from different reaction pathways and were used to underpin the proposed reaction scheme. This is an excellent, original study which was carefully conducted and evaluated. The proposed reaction schemes are reasonable and - wherever possible - supported with earlier findings by the authors and in the literature, although it has some speculative moments. But I see the latter as a challenge and I am wondering if the authors have ideas how a proof for the scheme in Fig 7 could look like. The paper is well written and very good to read (with a few exceptions, where formulations seem to be a little bit intricate). The results are original and new and give further deep insights into autoxidation and formation of highly oxidized molecules. This excellent manuscript should be published in ACP as it is.

**Reply:** We thank the reviewer for this comment. Unfortunately, the authors have no further idea how to proof the proposed reaction schemes in Figure 7.

Minor comments:
The authors may want to consider the following minor suggestions:

Figures are not addressed in sequence of their numberings in manuscript.

**Reply:** We addressed the figures now according to their numbering in the manuscript and added the following sentence:

**Changes in the text:** The number of oxygen atoms arising from the initial ozone attack was confirmed in experiments with isotopically labeled ozone ($^{18}O_3$) (Fig. 2).

Further possible reaction pathways of species 4b forming "norm. AutOx. and "ext. AutOx." reaction products are proposed in Fig. 6.

page 5, line 28: "Up to now" does not seem the right intro for what is following.

**Reply:** We removed the intro "up to now" from the sentence.

page 6, line 21: Maybe it is better use "analysis" instead of "investigation" in this context.

**Reply:** We changed the text accordingly.

page 6, line 26: I think it is better to talk about "HOM signal" instead of "HOM yield", the yield should be the same, independent of the detection scheme.

**Reply:** We changed the expression.

page 7, line 17: It may be simpler to replace "comparing by using" by "applying".

**Reply**: The term was changed accordingly.

page 7, line 19: typo, . . . second oxygen atom from the initial ozone attack must "have been" abstracted . . .

**Reply:** .We corrected this typo.

page 7, line 28: an acylalkoxy radical "(species 15)". Addition of species number would be helpful.

**Reply:** The species number 15 was added.

page 11, line 26: "The analysis has been done using nitrate ionization." may be better "These measurements were performed applying nitrate ionization" ?

**Reply:** The sentence was changed according to the reviewers suggestion.

page 12, line 10: typo, "of" missing.

**Reply:** Unfortunately, we did not see, where the "of" is missing.

Figure 2, caption: Although it is explained later in the caption it confused me that there was no blue signal shown in the Figure. May be better "(in black with blue label)" or so.

**Reply:** .We changed the figure caption accordingly

Supplement line 15: "deflected" seems to be more appropriate to me than "sucked".

**Reply:** We changed the expression.

We thank the reviewer for the helpful remarks on our manuscript. Please find our point-by-point responses below.

**Reviewers' comments to author:**

**Reviewer: 2**

Comments:
This is an interesting manuscript describing a continuation of the work from this laboratory on studying autoxidation reactions of biogenic terpenoids. The experiments seem carefully conducted with a previously described instrumentation that is especially suited for studying end product distributions of complex VOC oxidation reactions. However, number of conclusions derived in this paper, especially concerning "the extended autoxidation mechanism", seem somewhat hastily derived - or already reported. Thereby I'm not sure if the manuscript, as it currently stands, brings enough new insight to merit it's publication in ACP.
Most importantly, I do not see a need for "an extended autoxidation mechanism" for various reasons. Firstly, it is of obvious relevance what is understood as "the old mechanism". If the "old mechanism" is only thought of including peroxy radical isomerization + O2 addition steps, then I guess we could talk about a "new mechanism" at some level. However, the whole process is dependent on easily abstractable H-atoms and suitable molecular geometries enabling the abstractions – this is about what is clear at the moment. At the current stage it's not unambiguously clear what kind of steps are needed to progress the autoxidation chain to reach into the most highly-oxidized products in monoterpene oxidation (recently Kurten et al. 2015 suggested that bimolecular steps might be needed to advance the α-pinene oxidation). What seems intuitively clear, however, is that we're only beginning to understand the importance and the details of the autoxidation progression. So at this level it seems very preliminary to talk about "extended autoxidation" as we do not have a clear concise picture what constitutes the "normal autoxidation" in this context. In any case, probably you cannot really isolate the different pathways, but can account for branching between pathways under different conditions. So in my humble opinion, there is no need to bring up a "new extended autoxidation mechanism" – it is all the same autoxidation, just with a few additional steps.
Secondly, the pathways suggested to represent this "extended mechanism" constitute unambiguous reaction steps – the CO2 elimination and endoperoxide formation. Importantly, it does not seem to be possible to separate this CO2 elimination pathway from a CO-loss pathway brought up previously in autoxidation studies (e.g., Rissanen 2014, 2015, Mentel 2015). Both of these processes occur from acyl type radicals – the CO loss before O2 addition and the CO2 loss after the O2 addition. So while it is definitely worth to (and you should) discuss the potential of this type of reaction pathway, with current results it is impossible to be sure which type of dissociation process actually occurred. This should be made absolutely clear, and reference to papers where COloss was discussed should be given. Thirdly, the endoperoxide formation that is given as the explanation to account for the formation of "too less acidic hydrogens" (see D2O experiments) has already been suggested in exactly in the same context in previous literature concerning HOM formation (Rissanen et al. 2015, Kurten et al. 2015). So to sum up, it is hard to see the novelty of this paper especially as the main results of •-caryophyllene ozonolysis were already published previously (see Richters, et al. 2016). Hence, even though the manuscript appears to be generally well made, I cannot support its publication without severe changes in the interpretation of the mechanistic pathways and corresponding modifications for the manuscript text.

**Reply:** We thank the reviewer for the assessment of the manuscript. This main criticism is already answered in our "Short reply to RC 2".

Minor points:
Some of the terminology seem a bit awkward. In certain places it seems useful to label where the oxygens come from (i.e., O,O-formalism), but in most cases I think it only hinders the reading. So I would propose to stick with common CxHyOz formalism most of the time and then use the more difficult format where you are talking about the mechanism.

**Reply:** The authors prefer the use of the terminology. It allows always to illustrate the number of hydroperoxide moieties as well as the number of oxygen atoms arising from the initial ozone attack. For instance, the common CxHyOz formalism cannot distinguish between "norm. AutOx." and "ext. AutOx." reaction products, which have the same molecular formula. This differentiation is possible with the O,O-formalism and we think that it helps to understand the difference between the autoxidation mechanisms.

Can there be a different detection sensitivity for the peroxy radicals in comparison with the closed-shell products? It seems somewhat counterintuitive that the RO2 radicals could have such a long lifetimes under the present experimental conditions.

**Reply:** The reviewer is right that the detection sensitivity for $RO_2$ radicals might differ from the detection sensitivity of the closed-shell products. However, we cannot investigate this without having standard compounds for quantification. Standard compounds are not available for closed-shell products and it is not possible to synthesize the intermediate reaction products, $RO_2$ radicals. With our experimental setup we can almost suppress bimolecular reactions of $RO_2$ radicals and thus, the $RO_2$ radicals mainly react in rather slow unimolecular reaction pathways to form closed-shell products. The missing of bimolecular reactions can explain the rather long lifetimes of $RO_2$ radicals concerning bimolecular reactions such as with other $RO_2$ radicals under the given experimental conditions.

How was O3 handled? I assume that the "18O3" flow still contains about 95 to 99% of 18O2 (due to O3 generator generating efficiency) and thereby this could lead to significant difficulties in tracking the amount of O-atoms that are left from ozonolysis and do not result from secondary reactions after the initiation. This would be especially severe in trying to understand the contribution of different pathways (see Section 3.2.).

**Reply:** The reviewer is right that $^{18}O_3$ was produced by passing $^{18}O_2$, premixed in $N_2$, through and ozone generator (Thermo Environmental Instruments 49C). Hence, $^{18}O_2$ will still be present in the carrier gas. However, only 10 mL min$^{-1}$ of $^{18}O_2$ was used, first diluted in 4,99 L min$^{-1}$ of $N_2$ and later diluted in 95 L min$^{-1}$ purified air. Hence, 0.05% of the present oxygen in the carrier gas is supposed to be $^{18}O_2$. This very small amount should not influence the addition of $^{16}O_2$ to the alkyl radicals and thus, the isotopically labeled $^{18}O$ atoms will stem from the ozone attack at the double bond.

**Changes in the text:** The concentration of remaining $^{18}O_2$ in the carrier gas was about 0.05% of the total $O_2$ concentration. Hence, $^{18}O_2$ cannot compete with $^{16}O_2$ in the autoxidation steps. Thus, the isotopically labeled $^{18}O$ atoms will stem from the ozone attack at the double bond.

Page 3, Line 27: The use of CH3COOH is not mentioned. Also in Page 4 and line 13.

**Reply:** $CH_3COOH$ was used in the experiment in Figure 8 to scavenge stabilized Criegee intermediates (sCIs). This addition allows to investigate if sCIs take part in the HOM formation processes. This does not seem to the case. An additional sentence was added to Page 3, line 27.

**Changes in the text:** The addition of CH$_3$COOH was used to scavenge stabilized Criegee intermediates from the ozonolysis of β-caryophyllene (Beck et al., 2011; Neeb et al., 1996).

Page 4, Line 3: How was caryophyllene sampling done? Did the GC and PTR methods indicate any differences in determined concentrations?

**Reply:** β-caryophyllene was sampled using heated PEEK capillaries (100°C) which directly sample the center air flow from the experiment to the instrument. Only GC-FID was used for quantification. Here, the air flow was resampled in a sampling loop and then injected in the GC-FID. PTR-MS was only used to monitor the caryophyllene concentration, not to quantify it.

Page 4, line 20: What is meant by mass spectrometer setting?

**Reply:** By mass spectrometer settings we mean the applied voltages and flow conditions in the CI-MS. We added this information to the main text.

**Changes in the text:** The mass spectrometer settings (applied voltages and flow rates) as well as the approach applied for the determination of HOM concentrations are equal to those described in detail by Berndt et al. (2015b).

Page 5, Line 8. The number quoted is only the estimated concentration of highly oxidized RO2. How large do you assume is the pool of other radical species (e.g., how much are there less oxidized RO2s and HO2)?

**Reply:** Indeed, we can only state concentrations for RO$_2$ radicals and closed-shell products that can be detected using CI-APi-TOF mass spectrometry. Therefore, it is difficult to guess the concentrations of other RO$_2$ radicals. The concentration of HO$_2$ is supposed to be very low as very little OH radicals are formed in these reactions (total molar OH radical yield: 6% (Shu and Atkinson, 1994)). Furthermore, no reaction products from the bimolecular reaction of RO$_2$ radicals with HO$_2$ were detected (+ 1 nominal Th to the respective RO$_2$ radical) which supports that very low HO$_2$ concentrations are present.

Page 5, Lines 20-32: Similar H/D exchange behavior was observed and discussed in Rissanen, et al. 2015.

**Reply:** The reviewer is right that Rissanen et al. (2015) discusses this H/D exchange behavior from the ozonolysis of α-pinene. However, α-pinene does not contain a second double bond and thus, the H/D exchange by one nominal mass unit less than expected must arise from another reaction mechanism.

Page 6, Line 25-25: How can the yield change when changing ionization method? What you want to say is that the detection sensitivity varies between products and ionization methods. But what this means to the determined yields then? Does these yields then mean anything?

**Reply:** Indeed, the yield cannot change, but the apparent yield (detected yield) can change. With the following sentence "The change of the detection sensitivity for different HOMs (especially for those containing a single hydroperoxide moiety) leads to a different contribution of the individual product groups to the total molar HOM yield when changing from nitrate ionization to acetate ionization.", we hope, we clearly stated that the detection sensitivity varied between the reagent ions.
We still think that the relative contribution of the reaction products to the total molar yield mean something. Indeed, the numbers vary when changing from nitrate to acetate ionization, but they can show that the "norm. AutOx." reaction pathway is of minor importance for the

HOM formation from the ozonolysis of β-caryophyllene. We changed the text in the following way:

**Changes in the text:** Thus, the "norm. AutOx." group contributes with 29% to the total molar HOM yield when detecting with nitrate ionization and with 35% when detecting with acetate ionization. These values are based on the lower-limit concentration calculations and on the different detection sensitivities of the different reagent ions, which are depending e.g. on the number of hydroperoxide moieties in the molecule of interest. Hence, a quantitative statement concerning the contributions of the three reaction product groups is difficult. However, the two new product groups "ext. AutOx." and "ext. AutOx. $-CO_2$" are crucial for the explanation of HOM formation from the ozonolysis of β-caryophyllene.

Page 7 (and others): Be careful with Figure and Table numbering. Currently there are "Figs." and "Figures" which are not in numerical order.

**Reply:** Thanks for careful reading, the figures appear now in numerical order in the text.

Page 7, Line 19: Couldn't this be as well accounted for by a reaction in which 18OH (derived from VHP decomposition) starts the autoxidation sequence?

**Reply:** We conducted experiments in the presence of the OH scavenger propane and did not see a difference in the product spectrum. Furthermore, the OH radical yield from the ozonolysis of β-caryophyllene is very low (6% (Shu and Atkinson, 1994)) and should not influence the reaction spectrum. Thus, the influence of the OH-radical induced oxidation of β−caryophyllene is not supposed to play a role here.

Page 9, Line 10-15: How certain are you that the H/D exchange was 100% complete?

**Reply:** We added sufficient $D_2O$ that the H/D exchange from the acid-reagent ion clusters ($(HNO_3)_{1-2}NO_3^-$ and $(CH_3COOH)_{1-2}CH_3COO^-$) was complete. Furthermore, we performed the same H/D exchange experiments from the ozonolysis of cyclohexene and α-cedrene (two alkenes containing only one double bond). Here, the whole signals were shifted by the number of expected acidic H atoms. This shows that enough $D_2O$ was added to ensure a complete H/D exchange. Furthermore, we also increased the $D_2O$ concentration from 8 to 25 % r.h. and did not see a change in the intensities of the two signals that are separated in the presence of $D_2O$.

For example, in Rissanen et al. 2015 and incomplete H/D shift as seen in reagent ions was shown to result in partial H/D shifts in products too. So how accurate is the determination of the importance of different pathways, based on H/D shift?

**Reply:** As stated above, we made sure that enough $D_2O$ was added to the experiments to have a complete H/D exchange. This was made adding more $D_2O$ and by using other alkenes which contain only one double bond using the same experimental setup.

Page 10: Would make sense to change section 3.4. to 3.1. to improve the readability.

**Reply:** The authors agree that it might be challenging in some parts to understand the classification in three reaction product groups in 3.1. However, such an introduction is necessary to understand the experimental results and we prefer to show the experimental results before presenting the proposed reaction mechanisms.

Page 11, Line 11: Similar epoxide formation is well-known from atmospheric isoprene oxidation (e.g. Paulot et al. 2009)

**Reply:** Indeed, Paulot et al., 2009 proposed an epoxide formation. However, they proposed the attack of an O atom from a hydroperoxide moiety, releasing an OH radical. This proposed mechanism is different from the attack of a $RO_2$ radical under formation of an epoxide and an acylalkoxy radical.

Page 11, Line 16: RO2 + RO2 is usually considered as progression, not termination.

**Reply:** The reaction of an $RO_2$ radical with another $RO_2$ radical can either lead to progression, e.g. forming RO radicals, or to termination, e.g. forming a carbonyl compound and a hydroxyl moiety. Therefore, we exchange the word "termination" by "bimolecular" in this sentence.

Page 11, Line 27: Is this the first time nitrate ionization has been reported to see simple carboxylic acids?

**Reply:** In these experiments, the concentrations of $CH_3COOH$ are high enough to detect acetic acid as a nitrate cluster. For a concentration of $[CH_3COOH] = 10^{14}$ molecules cm$^{-3}$, a calculated concentration of $[(CH_3COOH)NO_3^-] = 10^7$ molecules cm$^{-3}$ was detected. This shows the very low sensitivity for these simple carboxylic acids.

Conclusions first sentence: Rather "end-product analysis was used to infer oxidation pathways".

**Reply:** We mainly investigated $RO_2$ radicals which are early intermediate reaction products and not "end products". We changed the text to the following sentence:

**Changes in the text:** Early reaction intermediates (mainly highly oxidized $RO_2$ radicals) from the ozonolysis of β-caryophyllene were investigated in a free-jet flow system at ambient pressure and a temperature of 295 ± 2 K to study the formation mechanisms of highly oxidized multifunctional organic compounds (HOMs).

Page 12, Line 26: I think the discussion on VOC sources etc. should be moved to discussion, after all it's not what was studied in this work.

**Reply:** We agree with the reviewer and deleted the sentences about VOC sources and the main oxidant from the conclusion.

Page 13, Line 1: "These, up to now undiscovered reaction pathways..." This sentence is an overstatement and simply not true. (In Line 10 a more appropriate wording is used).

**Reply:** We changed the sentence to the following:

**Changes in the text:** These new insights in $RO_2$ radical reaction pathways were investigated in a free-jet flow system...

Figure 7: I find it a bit odd that at the same time it's stated that the abstraction is "an example only" and then resulting species are said to be detected in the spectra. And the assumed epoxide structure seems questionable.

**Reply:** In the mass spectrometer, we can only detect the chemical composition of substances, the molecular structure in Figure 7 is just a proposed structure based on the

experimental results. Therefore, we cannot state which H atom was abstracted during the oxidation and hence, it represents an example only. However, we were able to detect a reaction product which contains the chemical composition, the number of acidic H atoms and the number of oxygen atoms from the initial ozone attack of the proposed molecule.

The epoxide structure is a proposed structure and we state ourselves that: "The epoxide formation cannot be proven and represents only a proposed reaction pathway in order to explain the experimental results."

We thank the reviewer for the helpful remarks on our manuscript. Please find our point-by-point responses below.

**Reviewers' comments to author:**

**Reviewer: 3**

Comments:
The manuscript, "Different Pathways..." by Richters et al. is well written, presents thorough experimental work and detailed spectral analyses that allowed the authors to identify three different pathways by which the Criegee intermediates formed by the ozonolysis of beta-caryophyllene undergo isomerization to become HOMs. The authors utilized a CI-API-ToF-MS with acetate as well as nitrate ionization coupled to a flow-tube. The use of isotopically labeled ozone and water vapor allowed robust identification of the ways in which the various RO2 peroxy radicals are formed, mainly, that a diene (presumably larger than a certain size) can undergo auto-oxidation in a few different ways to form unique molecular products. The branching ratios of these three pathways are also quantified. The work presented will be a significant contribution to the growing body of research on these HOM species, thus, should be published in ACP with some clarifications.

The D2O experiment (highlighted by figure 3 and SI figure 1) is key to differentiating "norm. AutOx." and "ext. AutOx." since the byproducts of each channel normally (in the absence of D2O) have the same molecular composition. The shift in mass by 1 amu between the y=1 and x=2 products (similarly, y = 2 and x=3) in the D2O experiment allows their distinction. How efficient or fast is the H/D exchange for these acidic H/D atoms? Is it possible that the y=1 peak is really "norm AutOx." product but with just one of the -OOH groups that has undergone H/D exchange?

**Reply:** The H/D exchange is always an equilibrium exchange and thus, we have to make sure that we add $D_2O$ in excess to enable a complete H/D exchange. We checked that by increasing the $D_2O$ concentration in the carrier gas to up to 25% relative humidity. No change in the signal intensities were visible when increasing from 8 to 25% r.h.. Furthermore, the reagent ions-acid clusters $((HNO_3)_{1-2}NO_3^-)$ are completely shifted to $((DNO_3)_{1-2}NO_3^-)$. Accordingly, $D_2O$ was added in excess and all acidic H atoms were exchanged by D atoms. We conducted the same experiments with cyclohexene (Berndt et al., 2015) and $\alpha$-cedrene (Richters et al., 2016). For these alkenes, which contain one double bond, the whole signals were shifted by the number of acidic H atoms assumed from the "normal" autoxidation mechanism. Hence, we can assume that the experimental conditions allow a complete H/D exchange and that the signal shift by one nominal mass unit less must arise from the missing of one acidic H atom giving reaction products from the "ext. AutOx." product group.

It appears that (in figures S1 and figure 3) that the red and blue peaks (y=1 and x=2) add up more-or-less to the corresponding black peak (no D2O). Would you expect this to be the case given that a compound with an endoperoxide group should have a different sensitivity (i.e. possess different ion cluster stability) compared to a compound with just hydroperoxide groups? The fact that red and blue add to up black would then suggest y=1 is really x=2 with one less H/D exchange.

**Reply:** The reviewer is right that HOMs with an endoperoxide moiety could be detected with a lower sensitivity than "norm. AutOx. HOMs. But we cannot prove or even quantify this. However, yes we would expect that the "norm. AutOx." (red) and "ext. AutOx." (blue) peaks add up to the corresponding black peak (no $D_2O$). If we assume that the sensitivity does not change when exchanging acidic H atoms with D atoms, we expect that the black signal must always represent the sum of the "red" and "blue" signal. If the black signal was only

composed of the "norm. AutOx." signal and the split-up was only due to an incomplete H/D exchange, the red and blue peaks should add up to the black peak. But this should also be the case, if the blue peak represents the "ext. AutOx." $RO_2$ radical. In the latter case, the black peak is composed of contributions from the "norm. AutOx" and the "ext. AutOx." product groups. Independent of the sensitivity, this signal should then be split up to the "red" and the "blue" signal and the intensities should add up to the intensity of the black signal. Furthermore, we checked that for HOMs from the ozonolysis of alkenes containing only one double bond, the signals are completely shifted by the expected number of D atoms (see answer above).

Given a reaction time of 7.9 seconds would you expect H/D exchange to be complete? Was there a time dependence experiment conducted (as in section 3.6) with D2O that demonstrates that the peak height for y=1 relative to x=2 does not change with residence or reaction time? If not, perhaps state in the manuscript that a peak where y=1 would reside was not observed for alpha-cedrene (monoalkene) in Richters et al [2016 ES&T figure 3]. Was [D2O] » [H2O] such that at equilibrium essentially all –OOH groups would be present as -OOD?

**Reply:** We did not conduct a time dependence experiment with $D_2O$ but changed the $D_2O$ concentration in the carrier gas. Here, no relative peak height changes between y = 1 and x = 2 were detectable. Thus, we expect that $D_2O$ was added in such an excess, that all acidic H atoms were exchanged by D atoms in this equilibrium reaction giving only –OOD groups. Indeed, the peak of y = 1 was not observed for $\alpha$-cedrene and this statement was added to the text as proposed by the reviewer.

**Changes in the text:** A signal which corresponds to the "ext. AutOx." $RO_2$ radical $O,O\text{-}C_{15}H_{23-y}(OO)(OOH)_yO_2$ with y = 1 was not detected as a product from the ozonolysis of $\alpha$-cedrene (a sesquiterpene with one double bond). Instead, the whole signal was shifted by the expected number of acidic H atoms, here two for $O,O\text{-}C_{15}H_{23-x}(OOH)_xO_2$ with x = 2. Accordingly, the concentration of heavy water was high enough to enable a complete exchange of acidic H atoms by D atoms.

As reported in lines 15-20 on page 6, the sensitivity (or ion cluster stability) difference between acetate and nitrate ionization to these HOMs (particularly those with one - OOH moiety) is quite large. It is reported (lines 21-30 page 6) that the "norm. AutOx" accounts for "between 29 and 35%" of HOM RO2 formation. This assertion is less than convincing. The two values do not really represent a range. It is two numbers from two different ionization schemes, each of which detects these compounds with varying efficiencies. There is a factor of 2-3 difference in these branching ratios for the "ext. AutOx" and "ext. AutoOx -CO2" pathways for the two ionization schemes. Given such large discrepancy, how is reliable or informative are the 29 and 35% numbers or any of these numbers? A discussion is needed on how the varying sensitivities of the ionization schemes are reconciled for this to be quantitative in any way. Moreover, though quantifying an absolute HOM yield is not the objective of this work, I find it uncomfortable that a single sensitivity value obtained from an inorganic acid (H2SO4) is applied to all of these multifunctional organic hydroperoxides and endoperoxides. Though this approach now has become routine, I strongly urge the authors to consider a more robust technique in the future to account for the varying sensitivities (depending on size, ring number, functional group, etc.) to these organics.

**Reply:** We agree with the reviewer that 29 and 35% are just two numbers and do not represent a range. Furthermore, all HOM concentrations represent lower limits and the determined concentrations using two different ionization techniques do only show that all reaction pathways are of importance. We changed the text in the following way:

**Changes in the text:** Thus, the "norm. AutOx." group contributes with 29% to the total molar HOM yield when detecting with nitrate ionization and with 35% when detecting with acetate ionization. These values are based on the lower-limit concentration calculations and on the different detection sensitivities of the different reagent ions, which are depending e.g. on the number of hydroperoxide moieties in the molecule of interest. Hence, a quantitative statement concerning the contributions of the three reaction product groups is difficult. However, the two new product groups "ext. AutOx." and "ext. AutOx. -$CO_2$" are crucial for the explanation of HOM formation from the ozonolysis of $\beta$-caryophyllene.

Furthermore, we agree that a single calibration factor should not be used to quantify all different HOMs. However, we decided to use this specific value only because it agreed with the calculations of the calculation factor. This calculation is described in detail in the literature (Berndt et al., 2015) and assumes an ideal and inlet system with a 12% diffusion loss in the inlet tube and a reaction time of 0.2-0.3 s (Berndt et al., 2015). The result of this calculation was a range of the calibration factor between (1.5 - 2.8) x $10^9$ molecule $cm^{-3}$. The calculation and the good agreement of the calculation with the experimental value (1.85 x $10^9$ molecule $cm^{-3}$) indicated that ionization reactions can be described properly. Therefore, we decided to use the experimental calibration factor, but this factor was checked using a general calculation.

Are the authors able to rule out pathways other than the three reported here? Are all peaks in the spectra accounted for by the three pathways? If not, what fraction of the observed peaks (at least the ones that can be reasonably identified as 1st generation products) are attributed to the three pathways? Is the alkoxy radical (RO dot) formation and subsequent degradation/isomerization relevant at all here?

**Reply:** With the three reported pathways, we are able to assign all intense signals in the mass spectra. There are additional small signals which we cannot assign, but these signals are of minor importance and do not influence the total molar HOM yield. The alkoxy radical formation needs bimolecular reaction mechanisms, these bimolecular reactions are almost completely suppressed within the setup and are not relevant here.

Please include a brief discussion on variables that may affect the branching ratios of the three pathways. Ambient pressure/temperature? Carbon number? The number of rings? Would MT or isoprene undergo "ext. AutoOx" and "ext. AutoOx. -CO2?

**Reply:** The reaction mechanisms were only investigated at one Temperature (T = 295 ± 2 K) and at ambient pressure. The influence of temperature and pressure was experimentally not studied.
All proposed reaction mechanisms are unimolecular reaction mechanisms. Here, a change in the pressure can only influence the rate coefficients if the reaction is not in its high-pressure limit. In our case, we cannot know if the investigated unimolecular reactions are in their high-pressure limit.
The unimolecular reactions have energy barriers. Thus, these reactions are supposed to be temperature dependent and the temperature dependence can have a strong influence on the specific unimolecular reaction and thus, on the branching ratios of the three pathways. However, we cannot state, which reaction pathway might be preferred when increasing the temperature.
The carbon number itself is not supposed to influence the rate coefficients however, structure and steric hindrance (and thus the number of rings) should influence the rate coefficients of the three reaction pathways. The number of weakly-bond H atoms in the precursor molecule and the presence of a second double bond are crucial for the importance of the pathways. For instance, the endoperoxide formation should be favored if five- or six-membered rings can be formed.

Accordingly, also monoterpenes should be able to undergo "ext. AutOx." and "ext. AutOx. -CO$_2$" if a second double bond is available or can be formed (e.g. for limonene). For isoprene, Vereecken and Peeters (2004) already proposed an endoperoxide formation starting from the OH radical initiated oxidation of isoprene.

We think that the discussion of a pressure, temperature, and structure dependence of the rate coefficients of the unimolecular reaction pathways is out of the scope of this manuscript, especially because we cannot provide experimental results for changing one of these variables. Thus, we did not add an additional paragraph to the manuscript and hope to answer the reviewer's question adequately.

In the future, the authors may want to consider not using the blue/red color combination since many have trouble distinguishing the two.

**Reply:** The reviewer is right that it can be hard to distinguish between these colors and the authors will consider a change in the future.

The term "acidic H atoms" is a bit vague. Please re-word in way that doesn't imply that these are H atoms from acid functional groups only. Perhaps "non-alkyl H atoms"?

**Reply:** The authors prefer to stay with "acidic H atoms" because this term does explain the chemical nature of these H atoms in the best way. Non-alkyl H atoms are not necessary acidic, e.g. if they are bond to another heteroatom.

Line 5 of page 9: "...three or two..." to "...three and two..."

**Reply:** We changed the text accordingly.

SI figure is really informative and deserves to be in manuscript not SI. Possible to combine with figure 3 or replace figure 3?

**Reply:** We agree with the reviewer and added the SI figure as Figure 4 to the main text.

[revised manuscript text omitted]